# KG-FIT: Knowledge Graph Fine-Tuning Upon Open-World Knowledge

**Pengcheng Jiang   Lang Cao   Cao Xiao**[†]  **Parminder Bhatia**[†]  **Jimeng Sun   Jiawei Han**

University of Illinois at Urbana-Champaign   [†]GE HealthCare

{pj20, langcao2, jimeng, hanj}@illinois.edu   danicaxiao@gmail.com

## Abstract

Knowledge Graph Embedding (KGE) techniques are crucial in learning compact representations of entities and relations within a knowledge graph, facilitating efficient reasoning and knowledge discovery. While existing methods typically focus either on training KGE models solely based on graph structure or fine-tuning pre-trained language models with classification data in KG, KG-FIT leverages LLM-guided refinement to construct a semantically coherent hierarchical structure of entity clusters. By incorporating this hierarchical knowledge along with textual information during the fine-tuning process, KG-FIT effectively captures both global semantics from the LLM and local semantics from the KG. Extensive experiments on the benchmark datasets FB15K-237, YAGO3-10, and PrimeKG demonstrate the superiority of KG-FIT over state-of-the-art pre-trained language model-based methods, achieving improvements of 14.4%, 13.5%, and 11.9% in the Hits@10 metric for the link prediction task, respectively. Furthermore, KG-FIT yields substantial performance gains of 12.6%, 6.7%, and 17.7% compared to the structure-based base models upon which it is built. These results highlight the effectiveness of KG-FIT in incorporating open-world knowledge from LLMs to significantly enhance the expressiveness and informativeness of KG embeddings.

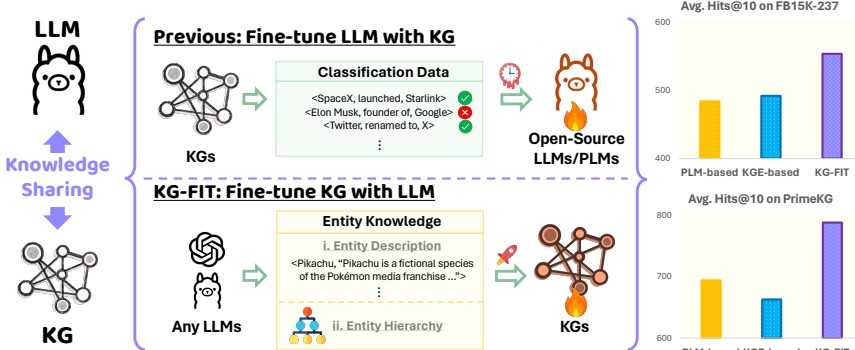

Figure 1: "Fine-tune LLM with KG" vs "Fine-tune KG with LLM".

## 1   Introduction

Knowledge graph (KG) is a powerful tool for representing and storing structured knowledge, with applications spanning a wide range of domains, such as question answering [1, 2, 3, 4], recommendation systems [5, 6, 7], drug discovery [8, 9, 10], and clinical prediction [11, 12, 13]. Constituted by entities and relations, KGs form a graph structure where nodes denote entities and edges represent relations among them. To facilitate efficient reasoning and knowledge discovery, knowledge graph embedding

38th Conference on Neural Information Processing Systems (NeurIPS 2024).

(KGE) methods [14, 15, 16, 17, 18, 19, 20, 21] have emerged, aiming to derive low-dimensional vector representations of entities and relations while preserving the graph's structural integrity.

While current KGE methods have shown success, many are limited to the graph structure alone, neglecting the wealth of open-world knowledge surrounding entities not explicitly depicted in the KG, which is manually created in most cases. This oversight inhibits their capacity to grasp the complete semantics of entities and relations, consequently resulting in suboptimal performance across downstream tasks. For instance, a KG might contain entities such as "Albert Einstein" and "Theory of Relativity", along with a relation connecting them. However, the KG may lack the rich context and background information about Einstein's life, his other scientific contributions, and the broader impact of his work. In contrast, pre-trained language models (PLMs) and LLMs, having been trained on extensive literature, can provide a more comprehensive understanding of Einstein and his legacy beyond the limited scope of the KG. While recent studies have explored fine-tuning PLMs with KG triples [22, 23, 24, 25, 26, 27, 28, 29], this approach is subject to several limitations. Firstly, the training and inference processes are computationally expensive due to the large number of parameters in PLMs, making it challenging to extend to more knowledgeable LLMs. Secondly, the fine-tuned PLMs heavily rely on the restricted knowledge captured by the KG embeddings, limiting their ability to fully leverage the extensive knowledge contained within the language models themselves. As a result, these approaches may not adequately capitalize on the potential of LLMs to enhance KG representations, as illustrated in Fig. 1. Lastly, small-scale PLMs (e.g., BERT) contain outdated and limited knowledge compared to modern LLMs, requiring re-training to incorporate new information, which hinders their ability to keep pace with the rapidly evolving nature of today's language models.

To address the limitations of current approaches, we propose `KG-FIT` (**K**nowledge **G**raph **FI**ne-**T**uning), a novel framework that directly incorporates the rich knowledge from LLMs into KG embeddings without the need for fine-tuning the LMs themselves. The term "fine-tuning" is used because the initial entity embeddings are from pre-trained LLMs, initially capturing global semantics.

`KG-FIT` employs a two-stage approach: (1) generating entity descriptions from the LLM and performing LLM-guided hierarchy construction to build a semantically coherent hierarchical structure of entities, and (2) fine-tuning the KG embeddings by integrating knowledge from both the hierarchical structure and textual embeddings, effectively merging the open-world knowledge captured by the LLM into the KG embeddings. This results in enriched representations that integrate both global knowledge from LLMs and local knowledge from KGs.

The main contributions of `KG-FIT` are outlined as follows:

(1) We introduce a method for automatically constructing a semantically coherent entity hierarchy using agglomerative clustering and LLM-guided refinement.

(2) We propose a fine-tuning approach that integrates knowledge from the hierarchical structure and pre-trained text embeddings of entities, enhancing KG embeddings by incorporating open-world knowledge captured by the LLM.

(3) Through an extensive empirical study on benchmark datasets, we demonstrate significant improvements in link prediction accuracy over state-of-the-art baselines.

## 2 Related Work

**Structure-Based Knowledge Graph Embedding.** Knowledge Graph Embedding methods that rely solely on graph structure aim to learn low-dimensional vector representations of entities and relations while preserving the graph's structural properties. TransE [14] models relations as translations in the embedding space. DistMult [15] is a simpler model that uses a bilinear formulation for link prediction. ComplEx [16] extends TransE to the complex domain, enabling the modeling of asymmetric relations. ConvE [17] employs a convolutional neural network to model interactions between entities and relations. TuckER [18] utilizes a Tucker decomposition to learn embeddings for entities and relations jointly. RotatE [19] represents relations as rotations in a complex space, which can capture various relation patterns, and HAKE [21] models entities and relations in an implicit hierarchical and polar coordinate system. These structure-based methods have proven effective in various tasks [30] but do not leverage the rich entity information available outside the KG itself.

**PLM-Based Knowledge Graph Embedding.** Recent studies have explored integrating pre-trained language models (PLMs) with knowledge graph embeddings to leverage the semantic information

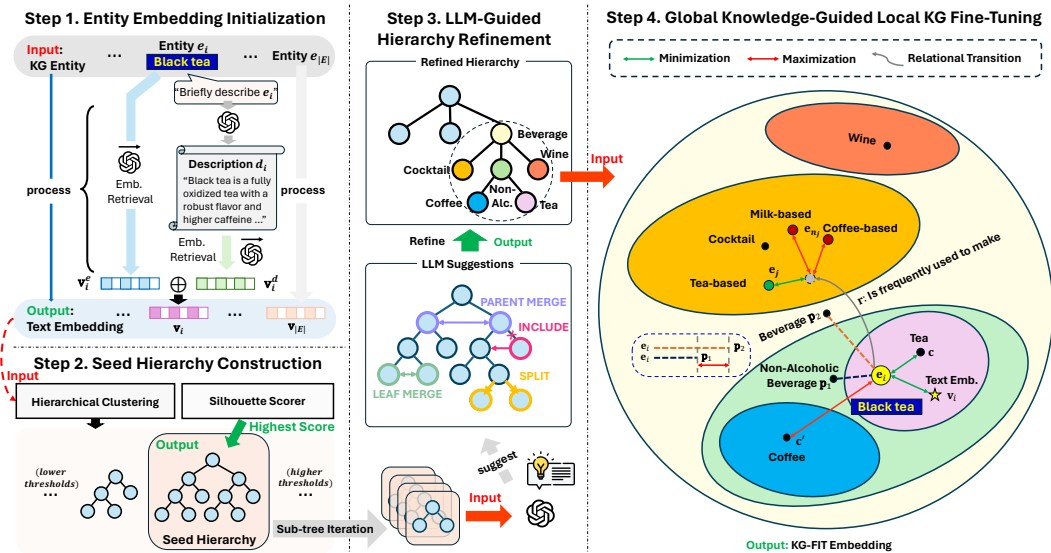

Figure 2: Overview of KG-FIT. **Input** and **Output** are highlighted at each step. **Step 1**: Obtain text embeddings for all entities in the KG, achieved by merging word embeddings with description embeddings retrieved from LLMs. **Step 2**: Hierarchical clustering is applied iteratively to all entity embeddings over various distance thresholds, monitored by a Silhouette scorer to identify optimal clusters, thus constructing a seed hierarchy where each leaf node represents a cluster of semantically similar entities. **Step 3**: Leveraging LLM guidance, the seed hierarchy is iteratively refined bottom-up through a series of suggested actions, aiming for a more accurate organization of KG entities with LLM's knowledge. **Step 4**: Use the refined hierarchy along with KG triples and the initial entity embeddings to fine-tune the embeddings under a series of distance constraints.

captured by PLMs. KG-BERT [22], PKGC [28], TagReal [29], and KG-LLM [31] train PLMs/LLMs with a full set of classification data and prompts. However, these approaches are computationally expensive due to the need to iterate over all possible positive/negative triples. LMKE [26] and SimKGC [27] adopt contrastive learning frameworks to tackle issues like expensive negative sampling and enable efficient learning for text-based KGC. KG-S2S [24] and KGT5 [25] employ sequence-to-sequence models to generate missing entities or relations in the KG. StAR [23] and CSProm-KG [32] fuse embeddings from graph-based models and PLMs. However, they are limited to small-scale PLMs and do not leverage the hierarchical and clustering information reflecting the LLM's knowledge of entities. Fully LLM prompting-based methods [33, 34] are costly and not scalable. In contrast, our proposed KG-FIT approach can be applied to any LLM, incorporating its knowledge through a semantically coherent hierarchical structure of entities. This enables efficient exploitation of the extensive knowledge within LLMs, while maintaining the efficiency of structure-based methods.

# 3 KG-FIT Framework

We present KG-FIT (as shown in Fig. 2), a framework for fine-tuning KG embeddings leveraging external hierarchical structures and textual information based on open knowledge. This framework comprises two primary components: (1) **LLM-Guided Hierarchy Construction**: This phase establishes a semantically coherent hierarchical structure of entities, initially constructing a seed hierarchy and then refining it using LLM-guided techniques, and (2) **Knowledge Graph Fine-Tuning**: This stage enhances the KG embeddings by integrating the constructed hierarchical structure, textual embeddings, and multiple constraints. The two stages combined to enrich KG embeddings with open-world knowledge, leading to more comprehensive and contextually rich representations. Below we present more tecnical details. A table of notations is placed in Appendix L.

## 3.1 LLM-Guided Hierarchy Construction

We initiate the process by constructing a hierarchical structure of entities through agglomerative clustering, subsequently refining it using an LLM to enhance semantic coherence and granularity.

**Step 1: Entity Embedding Initialization** is the first step of this process, where we are given a set of entities $\mathcal{E} = \{e_1, \ldots, e_{|\mathcal{E}|}\}$ within a KG, and will enrich their semantic representations by generating descriptions using an LLM. Specifically for each entity $e_i$, we prompt the LLM with a template (e.g., `Briefly describe [entity] with the format "[entity] is a [description]"`. (detailed in Appendix E.1)) prompting it to describe the entity from the KG dataset, thereby yielding a concise natural language description $d_i$. Subsequently, the entity embedding $\mathbf{v}_i^e \in \mathbb{R}^{\dim(f)}$ and description embedding $\mathbf{v}_i^d = f(d_i) \in \mathbb{R}^{\dim(f)}$ are obtained using an embedding model $f$ and concatenated to form the enriched entity representation $\mathbf{v}_i$:

$$\mathbf{v}_i = [\mathbf{v}_i^e; \mathbf{v}_i^d]. \tag{1}$$

**Step 2: Seed Hierarchy Construction** follows after entity embedding initialization. Here we choose agglomerative hierarchical clustering [35] over flat clustering methods like K-means [36] for establishing the initial hierarchy. This choice is based on the robust hierarchical information provided by agglomerative clustering, which serves as a strong foundation for LLM refinement. Using this hierarchical structure reduces the need for numerous LLM iterations to discern relationships between flat clusters, thereby lowering computational costs and complexity. Agglomerative clustering balances computational efficiency with providing the LLM a meaningful starting point for refinement. The clustering process operates on enriched entity representations $\mathbf{V} = \{\mathbf{v}_1, \ldots, \mathbf{v}_l\} \in \mathbb{R}^{|\mathcal{E}| \times 2\dim(f)}$ using cosine distance and average linkage. The optimal clustering threshold $\tau^*$ is determined by maximizing the silhouette score [37] $S^*$ across a range of thresholds $[\tau_{\min}, \tau_{\max}] \in [0, 1]$:

$$\tau_{\text{optim}} = \arg\max_{\tau \in [\tau_{\min}, \tau_{\max}]} S^*(\mathbf{V}, \text{labels}_\tau) \tag{2}$$

where $\text{labels}_\tau$ are the clustering results at threshold $\tau$. This ensures that the resulting clusters are compact and well-separated based on semantic similarity. The constructed hierarchy forms a fully binary tree where each leaf node represents an entity. We use a top-down algorithm (detailed in Appendix F.1) to replace the first encountered entity with its cluster based on the optimal threshold $\tau_{\text{optim}}$. This process eliminates other entity leaves within the same cluster, forming the seed hierarchy $\mathcal{H}_{\text{seed}}$, where each leaf node is a cluster of entities defined by $\text{labels}_{\tau_{\text{optim}}}$.

**Step 3: LLM-Guided Hierarchy Refinement (LHR)** is then applied to improve the quality of the knowledge representation. As the seed hierarchy $\mathcal{H}_{\text{seed}}$ is a binary tree, which may not optimally represent real-world entity knowledge, we further refine it using the LLM. The LLM transforms the seed hierarchy into the LLM-guided refined hierarchy $\mathcal{H}_{\text{LHR}}$ through actions described below:

***i. Cluster Splitting:*** For each leaf cluster $C_{\text{original}} \in \mathcal{C}_{\text{leaf}}(\mathcal{H}_{\text{seed}})$, the LLM recursively splits it into two subclusters using the prompt $\mathcal{P}_{\text{SPLIT}}$ (Fig. 8 in Appendix E.2):

$$C_{\text{split}} = \text{LLM}(\mathcal{P}_{\text{SPLIT}}(C_{\text{original}})), \quad C_{\text{original}} \to C_{\text{split}} = \{C_1, C_2, \ldots, C_k\}, \tag{3}$$

where $C_i = \{e_1^i, e_2^i, \ldots, e_{|C_i|}^i\}$, $\sum_{i=1}^k |C_i| = |C_{\text{original}}| = |C_{\text{split}}|$, $k$ is the total number of subclusters after recursively splitting $C_{\text{original}}$ in a binary manner. This procedure iterates until LLM indicates no further splitting or each subcluster has minimal entities, resulting in an intermediate hierarchy $\mathcal{H}_{\text{split}}$.

***ii. Bottom-Up Refinement:*** In the bottom-up refinement phase, the LLM iteratively refines the intermediate hierarchy $\mathcal{H}_{\text{split}}$ produced by the cluster splitting step. The refinement process starts from the leaf level of the hierarchy and progresses upwards, considering each parent-child triple $(P_*, P_l, P_r)$, where $P_*$ represents the grandparent cluster, and $P_l$ and $P_r$ represent the left and right child clusters of $P_*$, respectively. Let $\{C_1^l, C_2^l, C_3^l, \ldots, C_{|P_l|}^l\}$ denote the children of $P_l$, and $\{C_1^r, C_2^r, C_3^r, \ldots, C_{|P_r|}^r\}$ denote the children of $P_r$.

For each parent-child triple, the LLM is prompted with $\mathcal{P}_{\text{REFINE}}(P_*, P_l, P_r)$ (Fig. 9 in Appendix E.2), which provides the names and entities of the grandparent and child clusters. The LLM then suggests a refinement action to update the triple based on its understanding of the relationships between the clusters. The refinement options are:

1. **NO UPDATE**: The triple remains unchanged, i.e., $(P_*', P_l', P_r') = (P_*, P_l, P_r)$.
2. **PARENT MERGE**: All the children of $P_l$ and $P_r$ are merged into the grandparent cluster $P_*$, resulting in $P_*' = \{C_1^l, C_2^l, C_3^l, \ldots, C_{|P_l|}^l, C_1^r, C_2^r, C_3^r, \ldots, C_{|P_r|}^r\}$. The original child clusters $P_l$ and $P_r$ are removed from the hierarchy.
3. **LEAF MERGE**: $P_*' = \{e_1, e_2, \ldots, e_p\}, P_l' = \emptyset, P_r' = \emptyset$, where $\{e_1, e_2, \ldots, e_p\} = P_l \cup P_r$.

4. **INCLUDE**: One of the child clusters is absorbed into the other, while the grandparent cluster remains unchanged. This can happen in two ways:

- $P'_* = P'_l = \{C^l_1, C^l_2, C^l_3, \ldots, C^l_{|P_l|}, P_r\}$, and $P'_r = \emptyset$, or
- $P'_* = P'_r = \{P_l, C^r_1, C^r_2, C^r_3, \ldots, C^r_{|P_r|}\}$, and $P'_l = \emptyset$.

The LLM determines the most appropriate refinement action based on the semantic similarity and hierarchical relationships between the clusters. The refinement process continues bottom-up, iteratively updating the triples until the root of the hierarchy is reached. The resulting refined hierarchy is denoted as $\mathcal{H}_{\text{LHR}}$. We place more details of the process in Appendix E.2 and F.2.

## 3.2 Global Knowledge-Guided Local Knowledge Graph Fine-Tuning

**Step 4:** `KG-FIT` fine-tunes the knowledge graph embeddings by incorporating the hierarchical structure, text embeddings, and three main constraints: the hierarchical clustering constraint, text embedding deviation constraint, and link prediction objective.

**Initialization of Entity and Relation Embeddings:** To integrate the initial text embeddings ($\mathbf{v}_i \in \mathbb{R}^{\dim(f)}$) into the model, the entity embedding $\mathbf{e}_i \in \mathbb{R}^n$ is initialized as a linear combination of a random embedding $\mathbf{e}'_i \in \mathbb{R}^n$ and the sliced text embedding $\mathbf{v}'_i = [\mathbf{v}^e_i[:\frac{n}{2}]; \mathbf{v}^d_i[:\frac{n}{2}]] \in \mathbb{R}^n$. The relation embeddings, on the other hand, are initialized randomly:

$$\mathbf{e}_i = \rho \mathbf{e}'_i + (1-\rho)\mathbf{v}'_i, \quad \mathbf{r}_j \sim N(0, \psi^2) \tag{4}$$

where $\rho$ is a hyperparameter controlling the ratio between the random embedding and the sliced text embedding. $\mathbf{r}_j \in \mathbb{R}^m$ is the embedding of relation $j$, and $\psi$ is a hyperparameter controlling the standard deviation of the normal distribution $N$. This initialization ensures that the entity embeddings start close to their semantic descriptions but can still adapt to the structural information in the KG during training. The random initialization of relation embeddings allows the model to flexibly capture the structural information and patterns specific to the KG.

**Hierarchical Clustering Constraint:** The hierarchical constraint integrates the structure and relationships derived from the adaptive agglomerative clustering and LLM-guided refinement process. This optimization enhances the embeddings for hierarchical coherence and distinct semantic clarity. The revised constraint consists of three tailored components:

$$\mathcal{L}_{\text{hier}} = \sum_{e_i \in E} \Big( \underbrace{\lambda_1 d(\mathbf{e}_i, \mathbf{c})}_{\textit{Cluster Cohesion}} - \lambda_2 \underbrace{\sum_{C' \in \mathcal{S}_m(C)} \frac{d(\mathbf{e}_i, \mathbf{c}')}{|\mathcal{S}_m(C)|}}_{\textit{Inter-level Cluster Separation}} - \lambda_3 \underbrace{\sum_{j=1}^{h-1} \frac{\beta_j(d(\mathbf{e}_i, \mathbf{p}_{j+1}) - d(\mathbf{e}_i, \mathbf{p}_j))}{h-1}}_{\textit{Hierarchical Distance Maintenance}} \Big) \tag{5}$$

where: $e_i$ and $\mathbf{e}_i$ represent the entity and its embedding. $C$ is the cluster that entity $e_i$ belongs to. $\mathcal{S}_m(C)$ represents the set of neighbor clusters of $C$ where $m$ is the number of nearest neighbors (determined by lowest common ancestor (LCA) [38] in the hierarchy). $\mathbf{c}$ and $\mathbf{c}'$ denote the cluster embeddings of $C$ and $C'$, which is computed by averaging all the embedding of entities under them (i.e., $\mathbf{c} = \frac{1}{|C|} \sum_{e_i \in C} \mathbf{e}_i \in \mathbb{R}^n$). $\mathbf{p}_j$ and $\mathbf{p}_{j+1}$ are the embeddings of the parent nodes along the path from the entity (at depth $h$) to the root, indicating successive parent nodes in ascending order. Each parent node is computed by averaging the cluster embeddings under it ($\mathbf{p} = \frac{1}{|P|} \sum_{C_i \in P} \mathbf{c}_i \in \mathbb{R}^n$). $d(\cdot, \cdot)$ is the distance function used to measure distances between embeddings. As higher levels of abstraction encompass a broader range of concepts, and thus a strict maintenance of hierarchical distance may be less critical at these levels, we introduce $\beta_j = \beta_0 \cdot e^{-\phi j}$ where $\beta_0$ is the initial weight for the closest parent, typically a larger value, $\phi$ is the decay rate, a positive constant that dictates how rapidly the importance decreases. $\lambda_1$, $\lambda_2$, and $\lambda_3$ are hyperparameters. In Eq 5, **_Inter-level Cluster Separation_** aims to maximize the distance between an entity and neighbor clusters, enhancing the differentiation and reducing potential overlap in the embedding space. This separation ensures that entities are distinctly positioned relative to non-member clusters, promoting clearer semantic divisions. **_Hierarchical Distance Maintenance_** encourages the distance between an entity and its parent nodes to be proportional to their respective levels in the hierarchy, with larger distances for higher-level parent nodes. This reflects the increasing abstraction and decreasing specificity, aligning the embeddings with the hierarchical structure of the KG. **_Cluster Cohesion_** enhances intra-cluster similarity by minimizing the distance between an entity and its own cluster center, ensuring that entities within the same cluster are closely embedded, maintaining the integrity of clusters.

**Semantic Anchoring Constraint:** To preserve the semantic integrity of the embeddings, we introduce the semantic anchoring constraint, which is formulated as:

$$\mathcal{L}_{\text{anc}} = -\sum_{e_i \in \mathcal{E}} d(\mathbf{e}_i, \mathbf{v}'_i) \tag{6}$$

where $\mathcal{E}$ is the set of all entities, $\mathbf{e}_i$ is the fine-tuned embedding of entity $e_i$, $\mathbf{v}'_i$ is the sliced text embedding of entity $e_i$, and $d(\cdot, \cdot)$ is a distance function. This constraint is crucial for large clusters, where the diversity of entities may cause the fine-tuned embeddings to drift from their original semantic meanings. This is also important when dealing with sparse KGs, as the constraint helps prevent overfitting to the limited structural information available. By acting as a regularization term, it mitigates overfitting and enhances the robustness of the embeddings [39].

**Score Function-Based Fine-Tuning:** `KG-FIT` is a general framework applicable to existing KGE models [14, 15, 16, 17, 18, 19, 20]. These models learn low-dimensional vector representations of entities and relations in a KG, aiming to capture the semantic and structural information within the KG itself. In our work, we perform link prediction to enhance the model's ability to accurately predict relationships between entities within the KG. Its loss is defined as:

$$\mathcal{L}_{\text{link}} = -\sum_{(e_i, r, e_j) \in \mathcal{D}} \left( \log \sigma(\gamma - f_r(\mathbf{e}_i, \mathbf{e}_j)) - \frac{1}{|\mathcal{N}_j|} \sum_{n_j \in \mathcal{N}_j} \log \sigma(\gamma - f_r(\mathbf{e}_i, \mathbf{e}_{n_j})) \right) \tag{7}$$

where $\mathcal{D}$ is the set of all triples in the KG, $\sigma$ is sigmoid function. $f_r(\cdot, \cdot)$ is the scoring function (detailed in Appendix G and K) defined by the chosen KGE model that measures the compatibility between the head entity embedding $\mathbf{e}_i$ and the tail entity embedding $\mathbf{e}_j$ given the relation $r$, $\mathcal{N}_j$ is the set of negative tail entities sampled for the triple $(e_i, r, e_j)$, $\mathbf{e}_{n_j}$ is the embedding of the negative tail entity $n_j$, and $\gamma$ is a margin hyperparameter. The link prediction-based fine-tuning minimizes the scoring function for the true triples $(e_i, r, e_j)$ while maximizing the margin between the scores of true triples and negative triples $(e_i, r, n_j)$. This encourages the model to assign higher scores to positive (true) triples and lower scores to negative triples, thereby enriching the embeddings with the local semantics in KG.

**Training Objective:** The objective function of `KG-FIT` integrates three constraints:

$$\mathcal{L} = \zeta_1 \mathcal{L}_{\text{hier}} + \zeta_2 \mathcal{L}_{\text{anc}} + \zeta_3 \mathcal{L}_{\text{link}} \tag{8}$$

where $\zeta_1$, $\zeta_2$, and $\zeta_3$ are hyperparameters that assign weights to the constraints.

**Note**: During fine-tuning, the time complexity per epoch is $O((|\mathcal{E}| + |\mathcal{T}|) \cdot n)$ where $|\mathcal{E}|$ is the number of entities, $|\mathcal{T}|$ is the number of triples, and $n$ is the embedding dimension. In contrast, classic PLM-based methods [22, 28, 29, 23] have a time complexity of $O(|\mathcal{T}| \cdot L \cdot n_{\text{PLM}})$ per epoch during fine-tuning, where $L$ is the average sequence length and $n_{\text{PLM}}$ is the hidden dimension of the PLM. This is typically much higher than `KG-FIT`'s fine-tuning time complexity, as $|\mathcal{T}| \cdot L \gg (|\mathcal{E}| + |\mathcal{T}|)$.

## 4 Experiment

### 4.1 Experimental Setup

We describe our experimental setup as follows.

**Datasets**. We consider datasets that encompass various domains and sizes, ensuring comprehensive evaluation of the proposed model. Specifically, we consider three datasets: **(1) FB15K-237** [40] (*CC BY 4.0*) is a subset of Freebase [41], a large collaborative knowledge base, focusing on common knowledge; **(2) YAGO3-10**

Table 2: **Datasets statistics.** #Ent./#Rel: number of entities/relations. #Train/#Valid/#Test: number of triples contained in the training/validation/testing set.

| Dataset | #Ent. | #Rel. | #Train | #Valid | #Test |
|---|---|---|---|---|---|
| FB15k-237 | 14,541 | 237 | 272,115 | 17,535 | 20,466 |
| YAGO3-10 | 123,182 | 37 | 1,079,040 | 5,000 | 5,000 |
| PrimeKG | 10,344 | 11 | 100,000 | 3,000 | 3,000 |

[42] is a subset of YAGO [43] (*CC BY 4.0*), which is a large knowledge base derived from multiple sources including Wikipedia, WordNet, and GeoNames; **(3) PrimeKG** [44] (*CC0 1.0*) is a biomedical KG that integrates 20 biomedical resources, detailing 17,080 diseases through 4,050,249 relationships. Our study focuses on a subset of PrimeKG, extracting 106,000 triples from the whole set, with processing steps outlined in Appendix B. Table 2 shows the statistics of these datasets.

Table 1: **Link Prediction Performance Comparison**. Results are averaged values (of ten runs for FB15K-237/PrimeKG and of three runs for YAGO3-10) of head/tail entity predictions. Top-3 results for each metric are **highlighted**. "*" indicates the results taken from method's original paper. KG-FIT consistently outperforms both PLM-based models and structure-based base models across all datasets and metrics, demonstrating its effectiveness in incorporating open-world knowledge from LLMs for enhancing KG embeddings.

| | | FB15K-237 | | | | | YAGO3-10 | | | | | PrimeKG | | | | |
|---|---|---|---|---|---|---|---|---|---|---|---|---|---|---|---|---|
| **PLM-based Embedding Methods** | | | | | | | | | | | | | | | | |
| Model | PLM | MR | MRR | H@1 | H@5 | H@10 | MR | MRR | H@1 | H@5 | H@10 | MR | MRR | H@1 | H@5 | H@10 |
| KG-BERT [22]* | BERT | 153 | .245 | .158 | – | .420 | – | – | – | – | – | – | – | – | – | – |
| StAR [23]* | RoBERTa | **117** | .296 | .205 | – | .482 | – | – | – | – | – | – | – | – | – | – |
| PKGC [28] | RoBERTa | 184 | .342 | .236 | .441 | .525 | 1225 | .501 | .426 | .596 | .660 | 219 | .485 | .391 | .565 | .625 |
| C-LMKE [26]* | BERT | 141 | .306 | .218 | – | .484 | – | – | – | – | – | – | – | – | – | – |
| KGT5 [25]* | T5 | – | .276 | .210 | – | .414 | – | .426 | .368 | – | .528 | – | – | – | – | – |
| KG-S2S [24]* | T5 | – | .336 | .257 | – | .498 | – | – | – | – | – | – | – | – | – | – |
| SimKGC [27] | BERT | – | .336 | .249 | – | .511 | – | – | – | – | – | 168 | .527 | .524 | .679 | .742 |
| CSProm-KG [32] | BERT | – | .358 | .269 | – | .538 | 1145 | .488 | .451 | .624 | .675 | 157 | .540 | .492 | .652 | .745 |
| LLM Emb. (zero-shot) TE-3-S | | 2044 | .023 | .002 | .035 | .068 | 22741 | .009 | .000 | .016 | .024 | 5581 | .000 | .000 | .000 | .000 |
| LLM Emb. (zero-shot) TE-3-L | | 1818 | .030 | .004 | .048 | .085 | 18780 | .015 | .000 | .019 | .032 | 4297 | .001 | .000 | .000 | .000 |
| **Structure-based Embedding Methods** | | | | | | | | | | | | | | | | |
| Model | Frame | $\mathcal{H}$ | MRR | H@1 | H@5 | H@10 | MR | MRR | H@1 | H@5 | H@10 | MR | MRR | H@1 | H@5 | H@10 |
| TransE — Base [14] | | 233 | .287 | .192 | .389 | .478 | 1250 | .500 | .398 | .626 | .685 | 182 | .048 | .000 | .043 | .124 |
| TransE — KG-FIT Seed | | 142 | .345 | .242 | .457 | .547 | 952 | .520 | .429 | .638 | .700 | 80 | .298 | .000 | .315 | .516 |
| TransE — KG-FIT LHR | | 122 | **.362** | .264 | .478 | .568 | **529** | .544 | .463 | .650 | .705 | 69 | .334 | .000 | .342 | .536 |
| DisMult — Base [15] | | 283 | .260 | .163 | .349 | .437 | 5501 | .451 | .365 | .553 | .615 | 174 | .577 | .475 | .699 | .782 |
| DisMult — KG-FIT Seed | | 184 | .316 | .198 | .415 | .512 | 963 | .486 | .413 | .591 | .673 | 107 | .589 | .495 | .715 | .799 |
| DisMult — KG-FIT LHR | | 154 | .331 | .226 | .433 | .529 | 861 | .527 | .441 | .636 | .682 | 78 | .617 | .526 | .747 | .813 |
| ComplEx — Base [16] | | 347 | .252 | .161 | .344 | .439 | 6681 | .463 | .384 | .560 | .612 | 202 | .614 | .522 | .728 | .789 |
| ComplEx — KG-FIT Seed | | 201 | .325 | .223 | .436 | .523 | 997 | .491 | .422 | .603 | .669 | 94 | .638 | .548 | .767 | .823 |
| ComplEx — KG-FIT LHR | | 151 | .344 | .247 | .458 | .551 | 842 | .544 | .460 | .646 | .697 | 82 | **.651** | **.566** | **.772** | **.835** |
| ConvE — Base [17] | | 341 | .312 | .224 | .401 | .508 | 1105 | .529 | .451 | .619 | .673 | 144 | .516 | .456 | .645 | .760 |
| ConvE — KG-FIT Seed | | 181 | .318 | .237 | .411 | .521 | 912 | .535 | .455 | .628 | .685 | 93 | .627 | .534 | .757 | .812 |
| ConvE — KG-FIT LHR | | 177 | .318 | .241 | .415 | .525 | 885 | .541 | .461 | .647 | .695 | 72 | .648 | .547 | .767 | .824 |
| TuckER — Base [18] | | 363 | .320 | .230 | .417 | .505 | 1110 | .529 | .454 | .633 | .690 | 171 | .543 | .442 | .663 | .737 |
| TuckER — KG-FIT Seed | | 175 | .330 | .241 | .433 | .521 | 874 | .538 | .458 | .651 | .703 | 77 | .640 | .542 | .770 | .805 |
| TuckER — KG-FIT LHR | | 144 | .349 | .255 | .448 | .543 | 838 | .545 | **.466** | **.654** | .708 | 62 | .648 | .550 | **.779** | .820 |
| pRotatE — Base[19] | | 188 | .310 | .205 | .399 | .502 | 974 | .477 | .385 | .573 | .655 | 118 | .491 | .399 | .593 | .681 |
| pRotatE — KG-FIT Seed | | 160 | .355 | .257 | .461 | .558 | 910 | .525 | .436 | .622 | .693 | 75 | .635 | .538 | .745 | .809 |
| pRotatE — KG-FIT LHR | | **119** | **.371** | **.277** | **.483** | **.572** | 829 | **.550** | .464 | .648 | **.710** | 69 | **.649** | **.574** | **.779** | **.833** |
| RotatE — Base [19] | | 190 | .333 | .241 | .428 | .528 | 1620 | .495 | .402 | .550 | .670 | 57 | .539 | .447 | .646 | .727 |
| RotatE — KG-FIT Seed | | 141 | .354 | .261 | .464 | .555 | 790 | .529 | .440 | .643 | .708 | **46** | .622 | .517 | .740 | .805 |
| RotatE — KG-FIT LHR | | **120** | **.369** | **.274** | **.488** | **.570** | **744** | **.563** | **.475** | **.658** | **.722** | **34** | .645 | .532 | .758 | .817 |
| HAKE — Base [20] | | 184 | .344 | .247 | .435 | .538 | 1220 | 530 | .431 | .634 | .681 | 95 | .595 | .515 | .708 | .760 |
| HAKE — KG-FIT Seed | | 162 | .358 | .268 | .470 | .563 | 854 | .541 | .455 | .647 | .703 | 82 | .638 | .540 | .747 | .808 |
| HAKE — KG-FIT LHR | | 137 | **.362** | **.275** | **.485** | **.572** | **810** | **.568** | **.474** | **.662** | **.718** | **42** | **.682** | **.605** | **.785** | **.835** |

**Metrics.** Following previous works, we use Mean Rank (MR), Mean Reciprocal Rank (MRR), and Hits@N (H@N) to evaluate link prediction. MR measures the average rank of true entities, lower the better. MRR averages the reciprocal ranks of true entities, providing a normalized measure less sensitive to outliers. Hits@N measures the proportion of true entities in the top $N$ predictions.

**Baselines**. To benchmark the performance of our proposed model, we compared it against the state-of-the-art **PLM-based methods** including KG-BERT [22], StAR [23], PKGC [28], C-LMKE [26], KGT5 [25], KG-S2S [24], SimKGC [27], and CSProm-KG [32], and **structure-based methods** including TransE [14], DistMult [15], ComplEx [16], ConvE [17], TuckER [18], pRotatE [19], RotatE [19], and HAKE [20].

**Experimental Strategy:** For most PLM-based models, due to their high training cost, we use the results reported in their respective papers.. We reproduce PKGC [28] for all three datasets, SimKGC [27] and CSProm-KG [32] for PrimeKG. In addition, we evaluate the capabilities of LLM embeddings (TE-3-S/L: *text-embedding-3-small/large*) for zero-shot link prediction by ranking the cosine similarity between $(\mathbf{e}_i + \mathbf{r})$ and $\mathbf{e}_j$. For structure-based KGE models, we assess and present their best performance using optimal settings (shown in Table 12). For KG-FIT, we provide a detailed hyperparameter study in Table 13. We use OpenAI's GPT-4o as the LLM for entity description generation and for LLM-guided hierarchy refinement. *text-embedding-3-large* is used for entity embedding initialization, which preserves semantics with flexible embedding slicing [45]. We set $\tau_{\min} = 0.15$ and $\tau_{\max} = 0.85$ for seed hierarchy construction. The values of $\tau_{\text{optim}}$ for FB15K-237,

YAGO3-10, and PrimeKG are 0.52, 0.49, and 0.33, respectively. The statistics of the seed and LHR hierarchies are placed in Appendix F.3. For LCA, we designate the root as the grandparent (i.e., two levels above) source cluster node and set $m = 5$. We set $\rho = 0.5$, $\beta_0 = 1.2$, $\phi = 0.4$, and use cosine distance for fine-tuning. Filtered setting [14, 19] is applied for link prediction evaluation. Hyperparameter studies and computational cost are detailed in Appendix I and H, respectively.

## 4.2 Results

We conduct experiments to evaluate the performance on link prediction. The results in Table 1 are averaged values from multiple runs with random seeds: ten runs for FB15K-237 and PrimeKG, and three runs for YAGO3-10. These averages reflect the performance of head/tail entity predictions.

**Main Results.** Table 1 shows that our KG-FIT framework consistently outperforms state-of-the-art PLM-based and traditional structure-based models across all datasets and metrics. This highlights KG-FIT's effectiveness in leveraging LLMs to enhance KG embeddings. Specifically, KG-FIT$_{HAKE}$ surpasses CSProm-KG by 6.3% and HAKE by 6.1% on FB15K-237 in Hits@10; KG-FIT$_{RotatE}$ outperforms CS-PromKG by 7.0% and TuckER by 4.6% on YAGO3-10; KG-FIT$_{ComplEx}$ exceeds PKGC by 11.0% and ComplEx by 5.8% on PrimeKG. Additionally, with LLM-guided hierarchy refinement (LHR), KG-FIT achieves performance gains of 12.6%, 6.7%, and 17.8% compared to the base models, and 3.0%, 1.9%, and 2.2% compared to KG-FIT with seed hierarchy, on FB15-237, YAGO3-10, and PrimeKG, respectively. All these findings highlight the effectiveness of KG-FIT for significantly improving the quality of structure-based KG embeddings.

Figure 3 further illustrates the robustness of KG-FIT, showing superior validation performance across training steps compared to the corresponding base models, indicating its ability to fix both overfitting and underfitting issues of some structure-based KG embedding models.

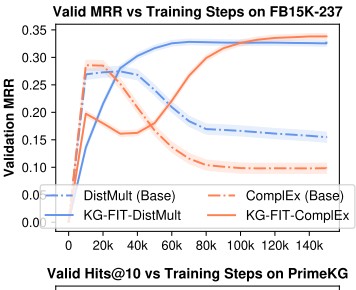

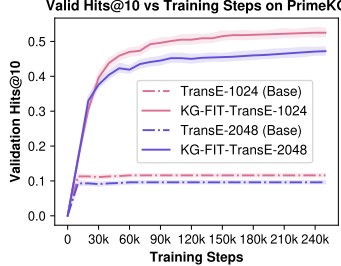

Figure 3: KG-FIT can mitigate overfitting (upper) and underfitting (lower) of structure-based models.

**Effect of Constraints.** We conduct an ablation study to evaluate the proposed constraints in Eq. 5 and 6, with results summarized in Table 4. This analysis underscores the importance of each constraint: **(1) Hierarchical Distance Maintenance** is crucial for both datasets. Its removal significantly degrades performance across all metrics, highlighting the necessity of preserving the hierarchical structure in the embedding space. **(2) Semantic Anchoring** proves more critical for the denser YAGO3-10 graph, where each cluster contains more entities, making it harder to distinguish between them based solely on cluster cohesion. The sparser FB15K-237 dataset is less impacted by the absence of this constraint. Similar to the semantics anchoring, the removal of **(3) Inter-level Cluster Separation** significantly affects the denser YAGO3-10 more than FB15K-237. Without this constraint, entities in YAGO3-10 may not be well-separated from other clusters, whereas FB15K-237 is less influenced. Interestingly, removing **(4) Cluster Cohesion** has a larger impact on the sparser FB15K-237 than on YAGO3-10. This difference suggests that sparse graphs rely more on the prior information provided by entity clusters, while denser graphs can learn this information more effectively from their abundant data.

**Effect of Knowledge Sources.** We explore the impact of the quality of LLM-refined hierarchies and pre-trained text embeddings on final performance, as illustrated in Figures 4 and 5. The results indicate that hierarchies constructed and text embeddings retrieved from more advanced LLMs consistently lead to improved performance. This finding underscores KG-FIT's capacity to leverage and evolve with ongoing advancements in LLMs, effectively utilizing the increasingly comprehensive entity knowledge captured by these models.

**Efficiency Evaluation.** We evaluate the efficiency performance in Table 3. While pure structure-based models are the fastest, our model significantly outperforms all PLM-based models in both

Table 3: Model efficiency on PrimeKG. T/Ep and Inf denote training time/epoch and inference time. KG-FIT outperforms all the PLM-based models.

| Method | LM | T/Ep | Inf |
|---|---|---|---|
| KG-BERT | RoBERTa | 170m | 2900m |
| PKGC | RoBERTa | 190m | 50m |
| TagReal | LUKE | 190m | 50m |
| StAR | RoBERTa | 125m | 30m |
| KG-S2S | T5 | 30m | 110m |
| SimKGC | BERT | 20m | 0.5m |
| CSProm-KG | BERT | 15m | 0.2m |
| KG-FIT (ours) | Any LLM | 1.2m | 0.1m |
| Structure-based | — | 0.2m | 0.1m |

Table 4: **Ablation study for the proposed constraints.** *SA*, *HDM*, *ICS*, *CC* denote Semantic Anchoring, Hierarchical Distance Maintenance, Inter-level Cluster Separation, and Cluster Cohesion, respectively. We use TransE and HAKE as the base models for `KG-FIT` on FB15K-237 and YAGO3-10, respectively.

| HDM | SA | ICS | CC | FB15K-237 (KG-FIT$_{TransE}$) | | | | YAGO3-10 (KG-FIT$_{HAKE}$) | | | |
|---|---|---|---|---|---|---|---|---|---|---|---|
| | | | | MRR | H@1 | H@5 | H@10 | MRR | H@1 | H@5 | H@10 |
| ✓ | ✓ | ✓ | ✓ | .362 | .264 | .478 | .568 | .568 | .474 | .662 | .718 |
| ✗ | ✓ | ✓ | ✓ | .345$_{(\downarrow.017)}$ | .248$_{(\downarrow.016)}$ | .454$_{(\downarrow.024)}$ | .542$_{(\downarrow.026)}$ | .558$_{(\downarrow.010)}$ | .467$_{(\downarrow.007)}$ | .654$_{(\downarrow.008)}$ | .709$_{(\downarrow.009)}$ |
| ✗ | ✗ | ✓ | ✓ | .335$_{(\downarrow.027)}$ | .241$_{(\downarrow.023)}$ | .444$_{(\downarrow.034)}$ | .533$_{(\downarrow.035)}$ | .545$_{(\downarrow.023)}$ | .452$_{(\downarrow.022)}$ | .640$_{(\downarrow.022)}$ | .695$_{(\downarrow.023)}$ |
| ✗ | ✓ | ✗ | ✓ | .343$_{(\downarrow.019)}$ | .244$_{(\downarrow.020)}$ | .449$_{(\downarrow.029)}$ | .538$_{(\downarrow.030)}$ | .544$_{(\downarrow.024)}$ | .453$_{(\downarrow.021)}$ | .643$_{(\downarrow.019)}$ | .691$_{(\downarrow.027)}$ |
| ✗ | ✓ | ✗ | ✗ | .332$_{(\downarrow.030)}$ | .239$_{(\downarrow.025)}$ | .437$_{(\downarrow.041)}$ | .529$_{(\downarrow.039)}$ | .558$_{(\downarrow.010)}$ | .465$_{(\downarrow.009)}$ | .656$_{(\downarrow.006)}$ | .711$_{(\downarrow.007)}$ |
| ✗ | ✗ | ✗ | ✗ | .287$_{(\downarrow.075)}$ | .192$_{(\downarrow.072)}$ | .389$_{(\downarrow.089)}$ | .478$_{(\downarrow.090)}$ | .530$_{(\downarrow.038)}$ | .431$_{(\downarrow.043)}$ | .634$_{(\downarrow.028)}$ | .681$_{(\downarrow.037)}$ |

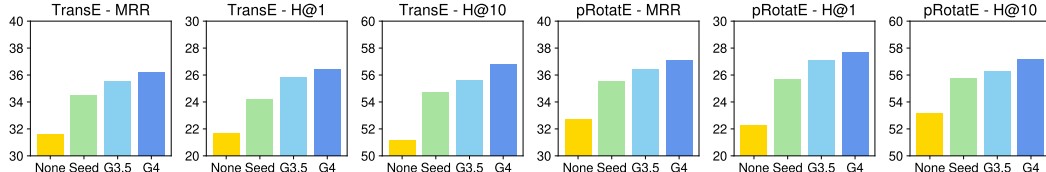

Figure 4: `KG-FIT` on FB15K-237 with different hierarchy types. *None* indicates no hierarchical information input. *Seed* denotes the seed hierarchy. *G3.5/G4* denotes the LHR hierarchy constructed by GPT-3.5/4o. LHR hierarchies outperform the seed hierarchy, with more advanced LLMs constructing higher-quality hierarchies.

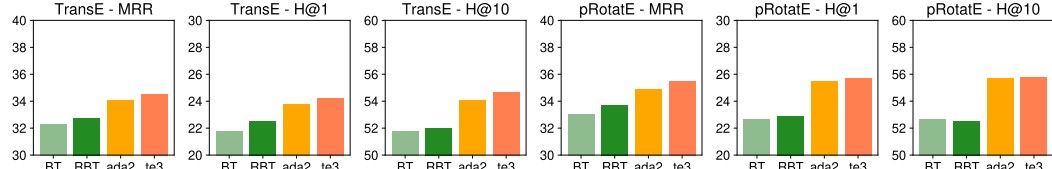

Figure 5: `KG-FIT` on FB15K-237 with different text embedding. *BT*, *RBT*, *ada2*, and *te3* are BERT, RoBERTa, text-embedding-ada-002, and text-embedding-3-large, respectively. Seed hierarchy is used for all settings. It is observed that pre-trained text embeddings from LLMs are substantially better than those from small PLMs.

training and inference speed, consistent with our previous analysis. It achieves 12 times the training speed of CSProm-KG, the fastest PLM-based method. Moreover, `KG-FIT` can integrate knowledge from any LLMs, unlike previous methods that are limited to small-scale PLMs, underscoring its superiority.

**Visualization.** Figure 6 demonstrates the effectiveness of `KG-FIT` in capturing both global and local semantics. The embeddings generated by `KG-FIT` successfully preserve the global semantics at both intra- and inter-levels. Additionally, `KG-FIT` excels in representing local semantics compared to the original HAKE model.

# 5   Conclusion

In this paper, we introduced `KG-FIT`, a novel framework for enhancing knowledge graph (KG) embeddings by leveraging the wealth of open-world knowledge captured by large language models (LLMs). `KG-FIT` seamlessly integrates LLM-derived entity knowledge into the KG embedding process through a two-stage approach: LLM-guided hierarchy construction and global knowledge-guided local KG fine-tuning. By constructing a semantically coherent hierarchical structure of entities and incorporating this hierarchical knowledge along with textual information during fine-tuning, `KG-FIT` effectively captures both global semantics from the LLM and local semantics from the KG. Extensive experiments on benchmark datasets demonstrate the superiority of `KG-FIT` over state-of-the-art methods, highlighting its effectiveness in integrating open-world knowledge from LLMs to significantly enhance the expressiveness and informativeness of KG embeddings. A key advantage of `KG-FIT` is its flexibility to incorporate knowledge from any LLM, enabling it to evolve and improve with ongoing advancements in language models. This positions `KG-FIT` as a powerful and future-proof framework for knowledge-infused learning on graphs. Moreover, the enriched KG embeddings produced by `KG-FIT` have the potential to boost performance on a wide array of downstream tasks, such as question answering, recommendation systems, and drug discovery, among others. Our code and data are available at https://github.com/pat-jj/KG-FIT.

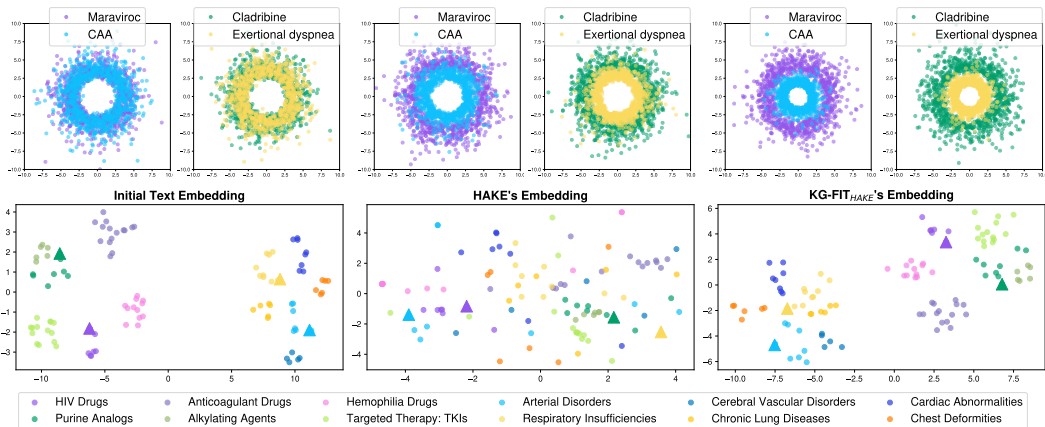

Figure 6: **Visualization of Entity Embedding (left to right: initial text embedding, HAKE embedding, and** KG-FIT$_{\textbf{HAKE}}$ **embedding).** *Upper (local)*: Embeddings (dim=2048) of <*Maraviroc, drug_effect, CAA (Coronary artery atherosclerosis)*> and <*Cladribine, drug_effect, Exertional dyspnea*>, two parent-child triples selected from PrimeKG, in polar coordinate system. In the polar coordinate system, the normalized entity embedding $\bar{\mathbf{e}}$ is split to $\mathbf{e_1} = \bar{\mathbf{e}}[: \frac{n}{2}]$ and $\mathbf{e_2} = \bar{\mathbf{e}}[\frac{n}{2} + 1 :]$ where $n$ is the hidden dimension, which serves as values on the x-axis and y-axis, respectively, which is consistent with Zhang et al. [20]'s visualization strategy. *Lower (global)*: t-SNE plots of different embeddings of sampled entities, with colors indicating clusters (e.g., *Maraviroc* belongs to the *HIV Drugs* cluster). Triangles indicate the positions of ▲ *Maraviroc*, ▲ *CAA*, ▲ *Cladribine*, and ▲ *Exertional dyspnea*. *Observations*: While the initial text embeddings capture global semantics, they fail to delineate local parent-child relationships within the KG, as seen in the intermingled polar plots. In contrast, HAKE shows more distinct grouping by modulus on the polar plots, capturing hierarchical local semantics, but fails to adequately capture global semantics. Our KG-FIT, notably, incorporates prior information from LLMs and is fine-tuned on the KG, maintains global semantics from pre-trained text embeddings while better capturing local KG semantics, demonstrating its superior representational power across local and global scales.

# 6 Limitations

Although KG-FIT outperforms state-of-the-art PLM-based models on the FB15K-237, YAGO3-10, and PrimeKG datasets, it does not outperform pure PLM-based methods on a lexical dataset WN18RR, as shown in Table 5. This limitation is discussed with details in Appendix C. As a future work, we will explore the integration of contrastive learning into the KG-FIT framework to enhance its capability to capture semantic relationships more effectively.

Moreover, KG-FIT's performance is influenced by the quality of the constructed hierarchy, particularly the seed hierarchy. To address this, we propose an automatic selection of the optimal binary tree based on the silhouette score. However, if the initial clustering is suboptimal, it may result in a lower-quality hierarchy that affects KG-FIT's performance. Additionally, the bottom-up refinement process in our proposed LHR approach updates each parent-child triple with only a single operation (within four), which prioritizes efficiency and simplicity over performance. In future work, we plan to explore cost-efficient methods for refining the hierarchy that integrate multiple operations for each triple update, striking a better balance between efficiency and performance.

# 7 Acknowledgement

The research was supported in part by US DARPA INCAS Program No. HR0011-21-C0165 and BRIES Program No. HR0011-24-3-0325, National Science Foundation IIS-19-56151, the Molecule Maker Lab Institute: An AI Research Institutes program supported by NSF under Award No. 2019897, and the Institute for Geospatial Understanding through an Integrative Discovery Environment (I-GUIDE) by NSF under Award No. 2118329. Any opinions, findings, and conclusions or recommendations expressed herein are those of the authors and do not necessarily represent the views, either expressed or implied, of DARPA or the U.S. Government.

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

**Contents of Appendix**

# A   Ethics and Broader Impacts

KG-FIT is a framework for enhancing knowledge graph embeddings by incorporating open knowledge from large language models. As a foundational research effort, KG-FIT itself does not have direct societal impacts. However, the enriched knowledge graph embeddings produced by KG-FIT could potentially be used in various downstream applications, such as question answering, recommendation systems, and drug discovery.

The broader impacts of KG-FIT depend on how the enhanced knowledge graph embeddings are used. On the positive side, KG-FIT could lead to more accurate and comprehensive knowledge representations, enabling better performance in beneficial applications like medical research and personalized recommendations. However, as with any powerful technology, there is also potential for misuse. The enriched knowledge could be used to build systems that spread disinformation or make unfair decisions that negatively impact specific groups.

To mitigate potential negative impacts, we encourage responsible deployment of KG-FIT and the enhanced knowledge graph embeddings it produces. This includes carefully monitoring downstream applications for fairness, robustness, and truthfulness. Additionally, when releasing KG-FIT-enhanced knowledge graph embeddings, we recommend providing usage guidelines and deploying safeguards to prevent misuse, such as gated access and safety filters, as appropriate.

As KG-FIT relies on large language models, it may inherit biases present in the language models' training data. Further research is needed to understand and mitigate such biases. We also encourage future work on building knowledge graph embeddings that are more inclusive and less biased.

Overall, while KG-FIT is a promising framework for advancing knowledge graph embeddings, it is important to consider its limitations and potential broader impacts. Responsible development and deployment will be key to realizing its benefits while mitigating risks. We are committed to fostering an open dialogue on these issues as KG-FIT and related technologies progress.

# B   PrimeKG Dataset Processing & Subset Construction

We describe the process of constructing a subset of the PrimeKG[1] [44] (version: V2, license: CC0 1.0) dataset. The goal is to create a highly-focused subset while leveraging additional information from the entire knowledge graph to assess KG embedding models' abilities on predicting drug-disease relationships.

The dataset construction process involves several detailed steps to ensure the creation of balanced training, validation, and testing sets from PrimeKG.

**Validation/Testing Set Creation:** We begin by selecting triples from the original PrimeKG where the type of the head entity is "drug" and the type of the tail entity is "disease." This selection yields a subset containing 42,631 triples, 2,579 entities, and 3 relations ("contraindication," "indication," "off-label use"). From this subset, we randomly select 3,000 triples each for the validation and testing sets, ensuring no overlap between the two.

**Training Set Creation:** We first extract the unique entities present in the validation and testing sets. These entities are used to ensure comprehensive coverage in the training set.

For each triple in the validation/testing set, we search for triples involving either its head or tail entity across the entire PrimeKG dataset, for the training set construction:

**Step 1** involves searching for 1-hop triples within the specified relations ("contraindication," "indication," "off-label use").

**Step 2** randomly enriches the training set with triples involving other relations, with a limit of up to 10 triples per entity.

**Step 3** involves removing redundant triples and any triples that are symmetric to those in the validation/testing set, enhancing the challenge posed by the dataset. If the training set contains fewer than 100,000 triples, we return to Step 2 and continue the process until the desired size is achieved.

---

[1]https://dataverse.harvard.edu/dataset.xhtml?persistentId=doi:10.7910/DVN/IXA7BM

Following our methodology, we construct a subset of the PrimeKG dataset that emphasizes drug-disease relations while incorporating broader context from the entire dataset. This subset is then split into training, validation, and testing sets, ensuring proper files are generated for training and evaluating knowledge graph embedding (KGE) models. The dataset contains **10,344 entities** with **11 types of relations**: `disease_phenotype_positive`, `drug_effect`, `indication`, `contraindication`, `disease_disease`, `disease_protein`, `disease_phenotype_negative`, `exposure_disease`, `drug_protein`, `off-label use`, `drug_drug`. The testing and validation sets include three specific relations ("`contraindication`," "`indication`," "`off-label use`").

## C   Results on WN18RR Dataset

Table 5: **Link Prediction Results on WN18RR**. Results are averaged values of ten independent runs of head/tail entity predictions. Top-6 results for each metric are highlighted in **bold.** "*" indicates the results taken from the method's original paper.

| WN18RR | | | | | | | |
|---|---|---|---|---|---|---|---|
| **PLM-based Embedding Methods** | | | | | | | |
| Model | | PLM | MR | MRR | H@1 | H@5 | H@10 |
| KG-BERT [22]* | | BERT | **97** | .216 | .041 | – | .524 |
| StAR [23]* | | RoBERTa | **51** | .401 | .243 | – | **.709** |
| PKGC [28] | | RoBERTa | 160 | .464 | .441 | .522 | .540 |
| C-LMKE [26]* | | BERT | **79** | **.619** | **.523** | – | **.789** |
| KGT5 [25]* | | T5 | – | .508 | .487 | – | .544 |
| KG-S2S [24]* | | T5 | – | .574 | **.531** | – | .661 |
| SimKGC [27]* | | BERT | – | **.666** | **.587** | – | **.800** |
| CSProm-KG [32]* | | BERT | – | .575 | **.522** | – | .678 |
| OpenAI-Emb (zero-shot) | | TE-3-S | 1330 | .141 | .075 | .205 | .271 |
| | | TE-3-L | 1797 | .127 | .074 | .178 | .235 |
| **Structure-based Embedding Methods** | | | | | | | |
| Model | Frame | $\mathcal{H}$ | MR | MRR | H@1 | H@5 | H@10 |
| pRotatE | Base[19] | — | 2924 | .460 | .416 | .506 | .552 |
| | `KG-FIT` | Seed | 165 | .566 | .498 | **.584** | .696 |
| | | LHR | **78** | **.590** | .511 | **.615** | **.722** |
| RotatE | Base [19] | — | 3365 | .476 | .429 | .523 | .572 |
| | `KG-FIT` | Seed | 144 | .554 | .501 | .571 | .712 |
| | | LHR | **75** | **.589** | .519 | **.600** | **.719** |
| HAKE | Base [20] | — | 3680 | .490 | .452 | .535 | .579 |
| | `KG-FIT` | Seed | 267 | .541 | .484 | **.590** | .660 |
| | | LHR | 172 | .553 | .488 | **.595** | .695 |
| **Combination** | | | | | | | |
| PKGC w/ `KG-FIT`$_{\text{pRotatE}}$'s recall ($\mathcal{X} = 20$) | | | **70** | **.625** | **.540** | **.655** | **.791** |
| `KG-FIT`$_{\text{pRotatE}}$ w/ SimKGC's Graph-Based Re-Ranking | | | **73** | **.624** | **.532** | **.652** | **.745** |

In this section, we present our results and findings on the WN18RR dataset. WN18RR [17] is derived from WordNet [46], a comprehensive lexical knowledge graph for the English language. This subset addresses the test leakage problems identified in WN18. The statistics of WN18RR are shown in Table 6. $\tau_{\text{optim}}$ we found for the seed hierarchy construction on WN18RR is 0.44.

Our experiments reveal that although `KG-FIT` did not surpass the performance of C-LMKE and SimKGC, its integration with PKGC [28], employing `KG-FIT` as a recall model followed by re-ranking with PKGC, yielded comparable results to these leading models.

Several factors contribute to these observations:

(1) WN18RR, being a lexical dataset, benefits significantly from pre-trained language model (PLM) based methods, which assimilate extensive lexical knowledge during fine-tuning. `KG-FIT`, relying on static knowledge graph embeddings, lacks this capability. However, as shown in Table 5, **its combination with PKGC leverages the strengths of both models, resulting in markedly improved performance.**

Table 6: **Statistics of WN18RR.** #Ent./#Rel: number of entities/relations. #Train/#Valid/#Test: number of triples contained in the set.

| Dataset | #Ent. | #Rel. | #Train | #Valid | #Test |
|---------|-------|-------|--------|--------|-------|
| WN18RR | 40,943 | 11 | 86,835 | 3,034 | 3,134 |

(2) **WN18RR has a very close semantic similarity between the head and tail entities**. The dataset is full of lexical relations such as "hypernym", "member_meronym", "verb_group", etc. **C-LMKE and SimKGC, both utilizing contrastive learning [47] with PLMs**, effectively exploit this semantic proximity during training. This ability to discern subtle semantic nuances contributes to their superior performance over other PLM-based approaches.

(3) **SimKGC implements a "Graph-based Re-ranking" strategy** that narrows the candidate pool to the k-hop neighbors of the source entity from the training set. This approach intuitively boosts performance by focusing on more relevant candidates. For a fair comparison, our experiments did not adopt this specific setting across all datasets. In Table 5, we showcase how this strategy could boost the performance for KG-FIT as well.

As a future work, we will explore the integration of contrastive learning into the KG-FIT framework to enhance its capability to capture semantic relationships more effectively.

## D   Supplemental Implementation Details

In this section, we provide more implementation details of both KG-FIT and baseline models, to improve the reproducibility of our work.

### D.1   KG-FIT

**Pre-computation.** In our KG-FIT framework, there is a pre-computation step to avoid overhead during the fine-tuning phase. The data we need to pre-compute includes:

1. Cluster embeddings $\mathbf{c}$: These are computed by averaging the all initial entity embeddings ($\mathbf{v}_i$) within each cluster.

2. Neighbor cluster IDs ($\mathcal{S}_m(C)$ in Eq. 5): These are computed using the lowest common ancestor (LCA) approach, where we set the ancestor as the grandparent (i.e., two levels above the node) and search for at most $m = 5$ neighbor clusters.

3. Parent node IDs: These represent the node IDs along the path from a cluster (leaf node) to the root of the hierarchy.

The pre-computed data is then used to efficiently locate the embeddings of clusters ($\mathbf{c}$, $\mathbf{c}'$) and parent nodes ($\mathbf{p}$) for the hierarchical clustering constraint during fine-tuning. With this pre-computation, we significantly speed up the training process, as the necessary information is readily available and does not need to be calculated on-the-fly.

**Distance Function.** We employ cosine distance as the distance metric for computing the hierarchical clustering constraint (Eq. 5) and the semantic anchoring constraint (Eq. 6). Cosine distance effectively captures the semantic similarity between embeddings, making it well-suited for maintaining the semantic meaning of the entity embeddings during fine-tuning. Cosine distance is also invariant to the magnitude of the embeddings, allowing the fine-tuned embeddings to adapt their magnitude based on the link prediction objective while preserving their semantic orientation with respect to the frozen reference embeddings. Moreover, cosine distance is independent of the link prediction objective, focusing on preserving the semantic properties of the embeddings.

Alternative distance metrics, such as Euclidean distance or L1 norm, were considered but deemed much slower in KG-FIT framework, aligning with the descriptions in OpenAI's document on the embedding models[2].

**Constraint Computation Options.** During fine-tuning, each training step involves a batch composed of one positive triple and $b$ negative samples, where $b$ is the negative sampling size. We provide two options for computing the constraints along with the link prediction score:

---

[2]https://platform.openai.com/docs/guides/embeddings/frequently-asked-questions

1. *Full Constraint Computation*: This option computes the distances for both the positive triple and the negative triples. For the positive triple, distances are computed for both the source and target entities. For the negative triples, distances are computed only for the source entities. The advantage of this option is that it allows the model to converge in fewer steps and generally achieves better performance after fine-tuning. However, the drawback is that the computation for each batch is nearly doubled, as positive and negative triples are separately fed into the model for score computation.

2. *Partial Constraint Computation*: This option computes the distances only for the source entities in the negative batch. It is faster than the full constraint computation option and can achieve comparable results based on our observations.

## D.2 Baseline Implementation

### D.2.1 PLM-based Methods

**PKGC.** For PKGC on FB15K-237, we use the templates the authors constructed and released [3] for FB15K-237-N, and use RoBERTa-Large as the base pre-trained langauge model. For YAGO3-10 and PrimeKG, we manually created templates for relations. For example, we converted the relation "isLeaderOf" to "[X] is a leader of [Y]." for YAGO3-10 and converted the relation "drug_effect" "drug [X] has effect [Y]". We choose TuckER as PKGC's backbone KGE recall model with hyperparameter $\mathcal{X} = 100$ for its overall great performance across all datasets. This means that we select top 50 results from TuckER and feed the shuffled entities into PKGC for re-ranking. We set batch size as 256 and run on 1 NVIDIA A6000 GPU for both training and testing.

**SimKGC.** For SimKGC on PrimeKG, for fairness, we do not use the "graph-based re-ranking" strategy introduced in their paper [27], which adds biases to the scores of known entities (in the training set) within $k$-hop of the source entity. The released code[4] was used for experiments. We set batch size as 256 and run on 1 NVIDIA A6000 GPU for both training and testing.

**CSProm-KG.** For CSProm-KG on PrimeKG, we use ConvE as the backbone graph model, which is the best Hits@N performed model. The other settings we use are the same as what reported in the paper. We use the code[5] released by the authors to conduct the experiments.

### D.2.2 Sturcture-based Methods

**TuckER and ConvE.** For TuckER, we use its release code[6] to run the experiments. For ConvE, we use its PyTorch implementation in PKGC's codebase[7], provided in the same framework as TuckER. It is worth noting that we do not use their proposed "label smoothing" setting for fair comparison.

**TransE, DistMult, ComplEx, pRotatE, RotatE.** For those KGE models, we reuse the code base[8] released by RotatE [19]. As it provides a unified framework and environment to run all those models, enabling a fair comparison. Our code ("code/model_common.py") also adapts this framework as the foundation.

**HAKE.** HAKE's code[9] was built upon RotatE's repository described above. In our work, we integrate the implementation of HAKE into our framework, which is also based on RotatE's.

The hyperparameters we used are presented in Appendix I.

```
# When hint is available
'''
Please provide a brief description of the entity '{entity}' in the following
format:
{entity} is a [description].

For example:
apple is a round fruit with red, green, or yellow skin and crisp, juicy flesh.

HINT:{hint}

Now, describe {entity}:
'''

# When there is no hint
'''
Please provide a brief description of the entity '{entity}' in the following
format:
{entity} is a [description].

For example:
apple is a round fruit with red, green, or yellow skin and crisp, juicy flesh.

Now, describe {entity}:
'''
```

Figure 7: Prompt for Entity Description.

# E    Prompts

## E.1    Prompt for Entity Description

Figure 7 showcases the prompts we used for entity description. In the prompt, we instruct the LLM
to provide a brief and concrete description of the input entity. We include an example, such as "apple
is a round fruit with red, green, or yellow skin and crisp, juicy flesh" to illustrate that the description
should be concise yet cover multiple aspects. Conciseness is crucial to minimize noise. For datasets
like FB15K-237 and WN18RR, where descriptions are already available [23], we use the original
description as a hint and ask the LLM to output the description in a unified format.

## E.2    Prompts for LLM-Guided Hierarchy Refinement

Fig. 8 and 9 show the main prompts we used for LLM-Guided Hierarchy Refinement (LHR).

**Prompt for Cluster Splitting** ($\mathcal{P}_{\textbf{SPLIT}}$): This prompt is designed to guide the LLM in identifying
and splitting a given cluster into meaningful subclusters based on the characteristics of the entities.
The prompt works as follows:

1. **Input Entities:** The entities from the specified cluster are provided as input.

2. **Analysis and Grouping:** The LLM is tasked with analyzing the entities to determine if they can
   be grouped into distinct subclusters based on common attributes like characteristics, themes, or
   genres.

---

[3]https://github.com/THU-KEG/PKGC
[4]https://github.com/intfloat/SimKGC
[5]https://github.com/chenchens190009/CSProm-KG
[6]https://github.com/ibalazevic/TuckER
[7]https://github.com/THU-KEG/PKGC/blob/main/TuckER/model.py
[8]https://github.com/DeepGraphLearning/KnowledgeGraphEmbedding
[9]https://github.com/MIRALab-USTC/KGE-HAKE

```
Given entities from the cluster '{cluster_name}'.

Analyze the entities and determine if they can be grouped into distinct and
meaningful sub-clusters based on their characteristics, themes, or genres.
If sub-clusters can be formed, provide a clear and concise name for each
sub-cluster that represents the common attribute of its entities.
Each sub-cluster should be given a new name that uniformly describes its
entities. There needs to be differentiation in the names of different
clusters.
The number of sub-clusters should be two.
If the entities are already well-grouped and don't require further sub-
clustering, simply provide the original cluster.

Provide the output in the following JSON format:
```json
{{
    "Sub-cluster 1 Name": ["Entity 1", "Entity 2", ...],
    "Sub-cluster 2 Name": ["Entity 3", "Entity 4", ...]
}}
```

Example:
Cluster: Movies
Entities: The Godfather, The Shawshank Redemption, The Dark Knight, Forrest
Gump, Inception, The Matrix
Output:
{{
    "Drama": ["The Godfather", "The Shawshank Redemption", "Forrest Gump"],
    "Action": ["The Dark Knight", "Inception", "The Matrix"]
}}

Cluster: {cluster_name}
Entities: {entities}
Output:
```

Figure 8: Prompt of LLM_SPLIT_CLUSTER.

3. **Sub-cluster Naming:** If subclusters are formed, the LLM provides clear and concise names for
   each subcluster that represent the common attributes of their entities. Each subcluster is given a
   unique name that uniformly describes its entities, ensuring differentiation between clusters.

4. **Control of Sub-cluster Count:** The prompt instructs the LLM to control the number of subclusters
   between 1 and 5. If the entities are already well-grouped, no further sub-clustering is needed, and
   the original cluster is returned.

The output format ensures structured and consistent results, facilitating easy integration into the
hierarchy refinement process.

**Prompt for Bottom-Up Refinement** ($\mathcal{P}_{\text{REFINE}}$): This prompt guides the LLM in refining the parent-
child triples within the hierarchy. The refinement process involves several key steps:

**1. Input Clusters:** Two clusters, A and B, along with their entities, are provided as input.

**2. Analysis of Clusters:** The LLM analyzes the two clusters and their entities to determine the most
appropriate update mode. The alignment of update modes to the actions described in the methodology
section is as follows:

- Update Mode 1 (Create New Cluster C): Aligns with NO UPDATE. These two clusters cannot be
  merged, and no cluster belongs to any other.

- Update Mode 2 (Merge Cluster A and B): Aligns with PARENT MERGE & LEAF MERGE. These
  two nodes can be merged. The name of the new merged node should be similar to both nodes.

```
Given the cluster A '{cluster_name_1}': [{entities_1}];
and the cluster B '{cluster_name_2}': [{entities_2}].

Analyze two clusters and their entities and determine the update mode:
Update Mode 1 - Create New Cluster C: these two clusters cannot be merged,
and no cluster belongs to any other.
Update Mode 2 - Merge Cluster A and B: these two clusters can be merged.
The name of two clusters should be similar and entities from two clusters
should be similar.
Update Mode 3 - Cluster A Covers Cluster B: cluster B belongs to cluster A.
cluster B is a subcluster of cluster A. The name of cluster A should
uniformly describe the entities from cluster A and the name of cluster B.
Update Mode 4 - Cluster B Covers Cluster A: cluster A belongs to cluster B.
cluster A is a subcluster of cluster B. The name of cluster B should
uniformly describe the entities from cluster B and the name of cluster A.

You need to select a update mode based on two clusters.
If you select mode 1, you should also suggest a name of new cluster. The
new cluster name should uniformly describe two clusters.
If you select mode 2, you should suggest a name of merged cluster. The new
name should be similar to cluster A and B.

Example:
Cluster A 'Thermal Insulators': [cork, fiberglass, foam];
Cluster B 'Electrical Conductors': [copper, aluminum, gold];
Select Mode 1.

Cluster A 'Sedans': [Toyota Camry, Honda Accord, Ford Fusion];
Cluster B 'SUVs': [Honda CR-V, Toyota RAV4, Ford Escape];
Select Mode 2.

Cluster A 'Feline Species': [lions, tigers, cheetahs];
Cluster B 'House Cats': [Siamese, Persian, Maine Coon];
Select Mode 3.

Cluster A 'Leafy Vegetables': [lettuce, spinach, kale];
Cluster B 'Root Vegetables': [carrots, potatoes, beets];
Select Mode 4.

Provide the output in the following JSON format:
```json
{{
    "update_mode": 1 or 2 or 3 or 4,
    "name": "merged cluster name or new cluster name"
}}
```

Output:
```

Figure 9: Prompt of LLM_UPDATE.

- Update Mode 3 & 4 (Cluster A Covers Cluster B): Aligns with INCLUDE where $P' = P \cup R, L' = L, R' = \emptyset$. Cluster B belongs to cluster A. Cluster B is a subcluster of cluster A. The name of cluster A should uniformly describe the entities from both clusters.

**3. Output Format:** The output includes the selected update mode and the suggested name for the new or merged cluster, ensuring clarity and consistency in the hierarchy refinement process.

These prompts, illustrated in Fig. 8 and Fig. 9, are essential for transforming the seed hierarchy into a more refined and accurate LLM-guided hierarchy, enabling better hierarchical representation to be learned by KG-FIT.

# F    Details of KG-FIT Hierarchy Construction

## F.1    Seed Hierarchy Construction

---

**Algorithm 1** Seed Hierarchy Construction

---

**Require:** Enriched entity representations $\mathbf{V}$, range of thresholds $[\tau_{\min}, \tau_{\max}]$
**Ensure:** Seed hierarchy $\mathcal{H}_{\text{seed}}$

  1:
  2: // *Step 1: Agglomerative Hierarchical Clustering*
  3: **for** $\tau \in [\tau_{\min}, \tau_{\max}]$ **do**
  4:       $\text{labels}_\tau \leftarrow \text{AgglomerativeClustering}(\mathbf{V}, \tau)$
  5:       $S_\tau \leftarrow \text{SilhouetteScore}(\mathbf{V}, \text{labels}_\tau)$
  6: **end for**
  7: $\tau_{\text{optim}} \leftarrow \arg\max(S_\tau)$
  8:
  9: // *Step 2: Constructing the Initial Hierarchy*
 10: $\mathcal{H}_{\text{init}} \leftarrow \text{ConstructBinaryTree}(\text{labels}_{\tau_{\text{optim}}})$
 11:
 12: // *Step 3: Top-Down Entity Replacement*
 13: $\text{visited\_clusters} \leftarrow \emptyset$
 14: $\text{ReplaceEntitiesWithClusters}(\mathcal{H}_{\text{init}}, \text{labels}_{\tau_{\text{optim}}}, \text{visited\_clusters})$
 15:
 16: // *Step 4: Refinement Steps*
 17: $\mathcal{H}_{\text{seed}} \leftarrow \text{Refine}(\mathcal{H}_{\text{init}})$
 18:
 19: **return** $\mathcal{H}_{\text{seed}}$
 20:
 21: **function** REPLACEENTITIESWITHCLUSTERS($\mathcal{H}$, labels, visited_clusters)
 22:       **for** node $\in \mathcal{H}$ **do**
 23:           **if** node is a leaf **then**
 24:               entity $\leftarrow$ GetEntity(node)
 25:               cluster $\leftarrow$ GetCluster(entity, labels)
 26:               **if** cluster $\in$ visited_clusters **then**
 27:                   Remove node from $\mathcal{H}$
 28:               **else**
 29:                   Replace node with cluster
 30:                   visited_clusters $\leftarrow$ visited_clusters $\cup$ {cluster}
 31:               **end if**
 32:           **else**
 33:               ReplaceEntitiesWithClusters(node, labels, visited_clusters)
 34:           **end if**
 35:       **end for**
 36: **end function**

---

This section details the seed hierarchy construction mentioned in Section 3.1. Algorithm 1 shows the pseudo code for this process.

The process begins with the agglomerative hierarchical clustering of the enriched entity representations $\mathbf{V}$ using a range of thresholds $[\tau_{\min}, \tau_{\max}]$. For each threshold $\tau$, the clustering labels ($\text{labels}_\tau$) are obtained, and the silhouette score ($S_\tau$) is calculated. The optimal threshold $\tau_{\text{optim}}$ is determined by selecting the threshold that maximizes the silhouette score.

Next, the initial hierarchy $\mathcal{H}_{\text{init}}$ is constructed based on the clustering labels obtained using the optimal threshold (labels$_{\tau_{\text{optim}}}$). The ConstructBinaryTree function builds a binary tree structure where each leaf node represents an entity.

The top-down entity replacement algorithm is then applied to the initial hierarchy $\mathcal{H}_{\text{init}}$. It traverses the hierarchy in a top-down manner, starting from the root node. For each node encountered during the traversal:

- If the node is a leaf, the entity associated with the node is retrieved using the GetEntity function.
- The cluster to which the entity belongs is obtained using the GetCluster function and the labels$_{\tau_{\text{optim}}}$.
- If the cluster has already been visited (i.e., it exists in the visited_clusters set), the leaf node is removed from the hierarchy.
- If the cluster has not been visited, the leaf node is replaced with the cluster, and the cluster is added to the visited_clusters set.
- If the node is not a leaf, the algorithm recursively applies the same process to its child nodes.

This top-down approach ensures that entities are included in their respective clusters as early as possible during the traversal of the hierarchy. By keeping track of the visited clusters, the algorithm avoids duplicating clusters in the hierarchy, resulting in a more compact and coherent representation.

Finally, the refinement steps (Refine function) are applied to the modified hierarchy to obtain the final seed hierarchy $\mathcal{H}_{\text{seed}}$. The refinement steps include handling empty dictionaries, single-entry dictionaries, and updating cluster assignments.

The resulting seed hierarchy $\mathcal{H}_{\text{seed}}$ represents a hierarchical organization of the entities, where each node is either a cluster containing a list of entities or a sub-hierarchy representing a more fine-grained grouping of entities. This seed hierarchy serves as a starting point for further refinement and incorporation of external knowledge in the subsequent steps of the KG-FIT framework.

## F.2 LLM-Guided Hierarchy Refinement

LLM-Guided Hierarchy Refinement (LHR) is used to further refine the seed hierarchy, which can better reflect relationships among entities. The process of LLM-Guided Hierarchy Refinement can be divided into two steps: LLM-Guided Cluster Splitting and LLM-Guided Bottom-Up Hierarchy Refinement.

As described in Algorithm 2, the LLM-Guided Cluster Splitting algorithm is designed to iteratively split clusters in a hierarchical structure with the guidance of a large language model (LLM). The algorithm begins with an initial hierarchy seed $H_{\text{seed}}$ and outputs a split hierarchy $H_{\text{split}}$. Initially, the current cluster ID is set to 0. The recursive procedure `RECURSION_SPLIT_CLUSTER` is then defined to manage the splitting process. If the root of the current cluster is a list and its length is less than a predefined minimum number of entities in a leaf node (`MIN_ENTITIES_IN_LEAF`), the function returns, indicating no further splitting is necessary. Otherwise, the root cluster's entities are passed to the LLM, which names the cluster and splits it into smaller clusters. If the split results in only one cluster, the function returns. For each resulting subcluster, the entities are assigned a new cluster ID, and the splitting process is recursively applied. If the root is not a list, the function iterates over each subcluster and applies the splitting procedure. Finally, the algorithm initiates the recursive splitting on the initial hierarchy seed and returns the modified hierarchy as $H_{\text{split}}$.

The LLM-Guided Bottom-Up Hierarchy Refinement algorithm (Algorithm 3) refines a previously split hierarchy $H_{\text{split}}$ to produce a more coherent and meaningful hierarchy $H_{\text{LHR}}$. This process is also guided by an LLM. The algorithm defines a recursive procedure `RECURSION_REFINE_HIERARCHY`, which starts by checking if the root of the current cluster is a list. If it is, the LLM is used to name the cluster, and the updated hierarchy is returned. Otherwise, the procedure recursively refines the left and right child nodes. The children of these nodes are then evaluated, and the LLM suggests an update mode based on the names and children of the right and left nodes. Depending on the suggested update mode, the algorithm may add the right node under the left children, the left node under the right children, or merge the left and right clusters. If no update is suggested, the procedure continues. The updated hierarchy is returned after applying the necessary refinements. The algorithm initiates the refinement process on the split hierarchy $H_{\text{split}}$ and outputs the refined hierarchy $H_{\text{LHR}}$.

**Algorithm 2** LLM-Guided Cluster Splitting
___

1: **Input:** $\mathcal{H}_{\text{seed}}$
2: **Output:** $\mathcal{H}_{\text{split}}$
3: current_cluster_id $\leftarrow$ 0
4: **procedure** RECURSION_SPLIT_CLUSTER(root)
5:     **if** root is a list **then**
6:         **if** length of root < MIN_ENTITIES_IN_LEAF **then**
7:             **return**
8:         **end if**
9:         cluster_entities $\leftarrow$ root
10:         cluster_name $\leftarrow$ LLM_NAME_CLUSTER(cluster_entities)
11:         splitted_clusters $\leftarrow$ LLM_SPLIT_CLUSTER(cluster_name, cluster_entities)
12:         **if** length of splitted_clusters == 1 **then**
13:             **return**
14:         **end if**
15:         **for** each (name, entities) in splitted_clusters **do**
16:             root[current_cluster_id] $\leftarrow$ entities
17:             current_cluster_id $\leftarrow$ current_cluster_id + 1
18:             RECURSION_SPLIT_CLUSTER(root[current_cluster_id])
19:         **end for**
20:     **else**
21:         **for** each (key, subcluster) in root **do**
22:             RECURSION_SPLIT_CLUSTER(subcluster)
23:         **end for**
24:     **end if**
25: **end procedure**
26: RECURSION_SPLIT_CLUSTER($\mathcal{H}_{\text{seed}}$)
27: $\mathcal{H}_{\text{split}} \leftarrow \mathcal{H}_{\text{seed}}$
28: **return** $\mathcal{H}_{\text{split}}$
___

### F.3    Statistics of Constructed Hierarchies

Table 7: Statistics of the hierarchies constructed by KG-FIT on different datasets.

| | | FB15K-237 | | YAGO3-10 | | PrimeKG | | WN18RR | |
|---|---|---|---|---|---|---|---|---|---|
| | | Seed | LHR | Seed | LHR | Seed | LHR | Seed | LHR |
| # Cluster | | 5,226 | 5,073 | 31,832 | 30,465 | 4,048 | 3,459 | 16,114 | 14,230 |
| # Node | | 10,452 | 8,987 | 63,664 | 52,751 | 8,096 | 5,918 | 32,228 | 26,186 |
| | Max | 115 | 115 | 1,839 | 1,839 | 72 | 72 | 40 | 58 |
| # Entity in a Cluster | Min | 1 | 1 | 1 | 1 | 1 | 1 | 1 | 1 |
| | Avg | 2.78 | 2.81 | 3.87 | 4.04 | 2.56 | 2.99 | 2.54 | 2.88 |
| | Max | 46 | 40 | 81 | 70 | 33 | 26 | 57 | 43 |
| Cluster Depth | Min | 3 | 3 | 4 | 4 | 4 | 3 | 5 | 3 |
| | Avg | 25.52 | 18.64 | 38.86 | 28.72 | 20.35 | 12.39 | 25.91 | 22.5 |
| | Max | 2 | 74 | 2 | 135 | 2 | 34 | 2 | 37 |
| # Branch of a Node | Min | 1 | 1 | 1 | 1 | 1 | 1 | 1 | 1 |
| | Avg | 2 | 2.3 | 2 | 2.37 | 2 | 2.41 | 2 | 2.19 |

Table 7 presents the statistics of the hierarchies constructed by KG-FIT on four different datasets: FB15K-237, YAGO3-10, PrimeKG, and WN18RR. The table compares the seed hierarchy (Seed) with the LLM-Guided Refined hierarchy (LHR) to show the changes and improvements after applying the LLM-guided hierarchy refinement process.

The number of clusters decreases across all datasets after refinement. For instance, in the FB15K-237 dataset, the clusters reduce from 5226 to 5073. Similarly, the number of nodes also decreases; for FB15K-237, nodes go from 10452 to 8987. The number of entities within each cluster sees a

---

**Algorithm 3** LLM-Guided Bottom-Up Hierarchy Refinement

---

1: **Input:** $\mathcal{H}_{\text{split}}$
2: **Output:** $\mathcal{H}_{\text{LHR}}$
3: **procedure** RECURSION_REFINE_HIERARCHY(root)
4:     **if** root is a list **then**
5:         updated_hierarchy ← LLM_NAME_CLUSTER(root)
6:         **return** updated_hierarchy
7:     **else**
8:         left_node ← RECURSION_REFINE_HIERARCHY(left_node)
9:         right_node ← RECURSION_REFINE_HIERARCHY(right_node)
10:         left_children ← Children(left_node)
11:         right_children ← Children(right_node)
12:         update_mode ← LLM_UPDATE(names and children of right and left node)
13:         **if** update_mode == no update **then**
14:             **continue**
15:         **else if** update_mode == merge left and right **then**
16:             Merge left and right clusters
17:             update hierarchy name and children
18:         **else if** update_mode == left include right **then**
19:             Add right_node under left_children
20:             update hierarchy name and children
21:         **else if** update_mode == right include left **then**
22:             Add left_node under right_children
23:             update hierarchy name and children
24:         **end if**
25:         **return** updated_hierarchy
26:     **end if**
27: **end procedure**
28: $\mathcal{H}_{\text{LRH}}$ ← RECURSION_REFINE_HIERARCHY($\mathcal{H}_{\text{split}}$)
29: **return** $\mathcal{H}_{\text{LHR}}$

---

slight increase in the average number, with the maximum and minimum values remaining fairly constant. For example, the average number of entities per cluster in FB15K-237 increases from 2.78 to 2.81. The depth of the hierarchies shows a noticeable reduction, with the maximum and average depths decreasing. In FB15K-237, the maximum depth goes from 46 to 40, and the average depth drops from 25.52 to 18.64. The branching factor of nodes also increases slightly, indicating a more interconnected structure; in FB15K-237, the average number of branches per node rises from 2 to 2.3. Similar patterns are observed in other datasets.

Overall, Table 7 illustrates that the refinement process effectively reduces the number of clusters and nodes, slightly increases the number of entities per cluster, decreases the depth of the hierarchy, and increases the branching factor. These changes suggest that the refinement process results in a more compact and interconnected hierarchy, better reflecting relationships among entities. The general effect of the refinement is the creation of a more streamlined and coherent hierarchical structure.

### F.4   Examples of LLM-Guided Hierarchy Refinement

Table 8: An example of LLM-Guided Cluster Splitting.

| Sub-cluster Name | Sub-cluster Entities |
|---|---|
| Universities and Colleges | Princeton University, Yale University, Harvard University, Brown University, Dartmouth College, Harvard College, Yale College |
| Medical Schools | Yale School of Medicine, Harvard Medical School |

Examples in Table 8 and Table 9 illustrate how the LLM-guided hierarchy refinement process updates and organizes clusters and sub-clusters to create a more coherent and meaningful hierarchical structure.

Table 8 provides examples of sub-clusters within a cluster in the process of cluster splitting. Each sub-cluster is given a name and lists the associated entities. For instance, one sub-cluster named

Table 9: Examples of LLM-Guided Bottom-Up Hierarchy Refinement.

| Case | Cluster A Name | Cluster A Entities | Cluster B Name | Cluster B Entities | Update |
|---|---|---|---|---|---|
| 1 | Washington D.C. Sports Teams | Washington Wizards, … | Detroit Sports Teams | Detroit Red Wings, … | No Update |
| 2 | New York Regional Entities | New York Island Entities, … | New York Locations and Cities | New York Locations, … | Parent Merge |
| 3 | Football Positions | running back, halfback, … | Football Positions | tight end, defensive end, … | Leaf Merge |
| 4 | Los Angeles Region | Los Angeles, Southern California | West Los Angeles Suburbs | Inglewood, Torrance | A Includes B |
| 5 | The Bryan Brothers | Bob Bryan, Mike Bryan | Tennis Athletes | Williams Sisters, … | B Includes A |

"Universities and Colleges" includes entities such as Princeton University and Harvard College. Additionally, a sub-cluster named "Medical Schools" includes entities such as Yale School of Medicine. The cluster of "Universities and Colleges" will be divided into the sub-clusters of "Universities" and "Colleges" in the next step. This example shows that LLM can divide the original large cluster into multiple smaller, refined ones.

Table 9 demonstrates various cases of cluster updates in LLM-Guided Bottom-Up Hierarchy Refinement. Each case lists two clusters (Cluster A and Cluster B) along with their entities and the type of update applied. In some cases, no update is needed, and both clusters remain unchanged. In others, Cluster B is merged into Cluster A, indicating a hierarchical relationship. There are instances where entities from Cluster B are integrated into Cluster A. Additionally, some updates involve Cluster A including all entities of Cluster B, making Cluster B a subset of Cluster A, or vice versa. These examples all show that the LLM correctly fixes the errors in the original seed hierarchy and refines the hierarchical structure to better reflect world knowledge.

Overall, these examples demonstrate how clusters are refined to better represent the relationships between entities, leading to a more organized and efficient structure. These examples highlight the detailed organization of entities into meaningful sub-clusters, reflecting their natural groupings and relationships. The refinement process not only improves the overall structure but also enhances the clarity and accessibility of the hierarchy.

## G   Score Functions

Table 10: Score functions defined by the KGE methods tested in this work.

| Model | Score Function $f_r(\mathbf{h}, \mathbf{t})$ | Parameters |
|---|---|---|
| TransE | $-\|\mathbf{h} + \mathbf{r} - \mathbf{t}\|_{1/2}$ | $\mathbf{h}, \mathbf{r}, \mathbf{t} \in \mathbb{R}^k$ |
| DistMult | $\mathbf{h}^\top \text{diag}(\mathbf{r})\mathbf{t}$ | $\mathbf{h}, \mathbf{r}, \mathbf{t} \in \mathbb{R}^k$ |
| ComplEx | $\text{Re}(\mathbf{h}^\top \text{diag}(\mathbf{r})\overline{\mathbf{t}})$ | $\mathbf{h}, \mathbf{r}, \mathbf{t} \in \mathbb{C}^k$ |
| ConvE | $f(\text{vec}(f(\overline{\mathbf{h}} * \omega))\mathbf{W})\mathbf{t}$ | $\mathbf{h}, \mathbf{r}, \mathbf{t} \in \mathbb{R}^k$ |
| TuckER | $\mathcal{W} \times_1 \mathbf{h} \times_2 \mathbf{w}_r \times_3 \mathbf{t}$ | $\mathbf{w}_r \in \mathbb{R}^{d_r}$ |
| pRotatE | $-2C \left\| \sin\left( \frac{\theta_h + \theta_r - \theta_t}{2} \right) \right\|_1$ | $\mathbf{h}, \mathbf{r}, \mathbf{t} \in \mathbb{C}^k$ |
| RotatE | $-\|\mathbf{h} \circ \mathbf{r} - \mathbf{t}\|_2$ | $\mathbf{h}, \mathbf{r}, \mathbf{t} \in \mathbb{C}^k, |\mathbf{r}_i| = 1$ |
| HAKE | $-\|\mathbf{h}_m \circ \mathbf{r}_m - \mathbf{t}_m\|_2 - \lambda\| \sin((\mathbf{h}_p + \mathbf{r}_p - \mathbf{t}_p)/2)\|_1$ | $\mathbf{h}_m, \mathbf{t}_m \in \mathbb{R}^k, \mathbf{t}_m \in \mathbb{R}^k_+,$ $\mathbf{h}_p, \mathbf{r}_p, \mathbf{t}_p \in [0, 2\pi)^k, \lambda \in \mathbb{R}$ |

The score functions defined by the structure-based KG embedding methods [14, 15, 16, 17, 18, 19, 20, 21] we tested in this papaer are shown in Table 10. Here is a paragraph explaining each notation in the table, with all explanations inline:

The table presents the score functions $f_r(\mathbf{h}, \mathbf{t})$ used by various knowledge graph embedding (KGE) methods, where $\mathbf{h}$, $\mathbf{r}$, and $\mathbf{t}$ represent the head entity, relation, and tail entity embeddings, respectively. TransE uses a translation-based score function with either L1 or L2 norm, denoted by $\| \cdot \|_{1/2}$, where the embeddings are in real space $\mathbb{R}^k$. DistMult employs a bilinear score function with a diagonal relation matrix $\text{diag}(\mathbf{r})$, and the embeddings are also in $\mathbb{R}^k$. ComplEx extends DistMult by using complex-valued embeddings in $\mathbb{C}^k$ and takes the real part of the score, denoted by $\text{Re}(\cdot)$. ConvE applies a 2D convolution operation, where $\overline{\mathbf{h}}$ is the 2D reshaping of $\mathbf{h}$, $*$ represents the

convolution, $\omega$ is a set of filters, $\text{vec}(\cdot)$ is a vectorization operation, and $\mathbf{W}$ is a linear transformation matrix. TuckER uses a Tucker decomposition with a core tensor $\mathcal{W}$ and relation-specific weights $\mathbf{w}_r \in \mathbb{R}^{d_r}$, where $\times_n$ denotes the tensor product along the $n$-th mode. pRotatE and RotatE employ rotation-based score functions in complex space, with $\circ$ representing the Hadamard product and $|\mathbf{r}_i| = 1$ constraining the relation embeddings to have unit modulus. HAKE combines a modulus part ($\mathbf{h}_m, \mathbf{t}_m \in \mathbb{R}^k, \mathbf{t}_m \in \mathbb{R}^k_+$) and a phase part ($\mathbf{h}_p, \mathbf{r}_p, \mathbf{t}_p \in [0, 2\pi)^k$) in its score function, with a hyperparameter $\lambda \in \mathbb{R}$ balancing the two parts.

# H    Computational Cost

## H.1    Hardware and Software Configuration

Based on the dataset size, we hybridly use two machines:

For FB15K-237, PrimeKG, and WN18RR, experiments are conducted on a machine equipped with two AMD EPYC 7513 32-Core Processors, 528GB RAM, eight NVIDIA RTX A6000 GPUs, and CUDA 12.4 and the NVIDIA driver version 550.76.

For YAGO3-10, due to its large size, experiments are conducted on a machine equipped with two AMD EPYC 7513 32-Core Processors, 528GB RAM, and eight NVIDIA A100 80GB PCIe GPUs. The system uses CUDA 12.2 and the NVIDIA driver version 535.129.03.

With a single GPU, it takes about 2.5, 4.5, 2.0, and 1.1 hours for a `KG-FIT` model to achieve good performance on FB15K-237, YAGO3-10, PrimeKG, and WN18RR, respectively.

## H.2    Cost of Close-Source LLM APIs

The costs of GPT-4o for **entity description generation** are $3.0, $24.8, $2.1, and $8.6 for FB15K-237, YAGO3-10, PrimeKG, and WN18RR, respectively, proportional to their numbers of entities.

The cost of text embedding models (text-embedding-3-large) for **entity embedding initialization** was totally about $0.8 to process all the KG datasets.

The costs of GPT-4o for **LLM-Guided Hierarchy Refinement** on FB15K-237, YAGO3-10, PrimeKG, and WN18RR are $10.0, $74.5, $8.7, and $34.4, respectively. This cost is almost proportional to the nodes consisted in the seed hierarchy.

# I    Hyperparameter Study

This section presents a comprehensive hyperparameter study for both structure-based base models and our proposed `KG-FIT` framework across different datasets. Table 11 outlines the range of hyperparameter values explored during the study. Table 12 showcases the optimal hyperparameter configurations for the base models that yielded the best performance. Similarly, Table 13 presents the best-performing hyperparameter settings for `KG-FIT`.

The hyperparameter study aims to provide insights into the sensitivity of the models to various hyperparameters and to identify the optimal configurations that maximize their performance on each dataset. By conducting a thorough exploration of the hyperparameter space, we ensure a fair comparison between the base models and `KG-FIT`, and demonstrate the robustness and effectiveness of our proposed framework across different settings.

Table 11: Summary of hyperparameters we explored for both base models and `KG-FIT`.

| Hyper-parameter | Studied Values |
|---|---|
| Batch Size | {64, 128, 256, 1024} |
| Negative Sampling Size ($|\mathcal{N}_j|$ in Eq 7) | {64, 128, 256, 400, 512, 1024} |
| Hidden Dimension Size $n$ | {512, 768, 1024, 2048} |
| Score Margin $\gamma$ | {6.0, 9.0, 10.0, 12.0, 24.0, 60.0, 200.0} |
| Learning Rate | {5e-5, 2e-4, 1e-4, 5e-4, 1e-3, 2e-3, 4e-3, 5e-3, 1e-2, 2e-2, 5e-2} |
| $\lambda_1, \lambda_2, \lambda_3$ | in range of [0.1, 1.0] |
| $\zeta_1, \zeta_2, \zeta_3$ | in range of [0.1, 10.0] |

Table 12: Best hyperparameters grid-searched for base models on different datasets.

**FB15K-237**

|  | TransE | DistMult | ComplEx | ConvE | TuckER | pRotatE | RotatE | HAKE |
|---|---|---|---|---|---|---|---|---|
| Batch Size | 1024 | 1024 | 1024 | 512 | 512 | 1024 | 1024 | 1024 |
| Learning Rate | 5e-4 | 1e-2 | 1e-2 | 5e-3 | 5e-3 | 5e-4 | 5e-4 | 1e-3 |
| Negative Sampling | 256 | 256 | 256 | – | – | 256 | 256 | 256 |
| Hidden Dimension | 1024 | 2048 | 1024 | 512 | 1024 | 2048 | 2048 | 2048 |
| $\gamma$ | 9.0 | 200.0 | 200.0 | – | – | 9.0 | 9.0 | 9.0 |

**YAGO3-10**

|  | TransE | DistMult | ComplEx | ConvE | TuckER | pRotatE | RotatE | HAKE |
|---|---|---|---|---|---|---|---|---|
| Batch Size | 512 | 512 | 512 | 128 | 128 | 256 | 1024 | 1024 |
| Learning Rate | 2e-3 | 1e-3 | 1e-4 | 5e-5 | 1e-4 | 5e-4 | 2e-3 | 2e-3 |
| Negative Sampling | 256 | 256 | 256 | – | – | 512 | 400 | 256 |
| Hidden Dimension | 1024 | 1024 | 1024 | 512 | 1024 | 1024 | 1024 | 1024 |
| $\gamma$ | 24.0 | 24.0 | 24.0 | – | – | 24.0 | 24.0 | 24.0 |

**PrimeKG**

|  | TransE | DistMult | ComplEx | ConvE | TuckER | pRotatE | RotatE | HAKE |
|---|---|---|---|---|---|---|---|---|
| Batch Size | 512 | 512 | 512 | 128 | 128 | 512 | 512 | 512 |
| Learning Rate | 5e-4 | 2e-2 | 2e-3 | 5e-4 | 5e-5 | 1e-4 | 1e-4 | 1e-4 |
| Negative Sampling | 512 | 512 | 512 | – | – | 1024 | 1024 | 512 |
| Hidden Dimension | 1024 | 1024 | 1024 | 512 | 1024 | 2048 | 2048 | 2048 |
| $\gamma$ | 24.0 | 200.0 | 200.0 | – | – | 24.0 | 24.0 | 6.0 |

Table 13: Hyperparameters we used for `KG-FIT` with different base models on different datasets.

**FB15K-237**

|  | TransE | DistMult | ComplEx | ConvE | TuckER | pRotatE | RotatE | HAKE |
|---|---|---|---|---|---|---|---|---|
| Batch Size | 512 | 512 | 512 | 256 | 256 | 256 | 256 | 256 |
| Learning Rate | 1e-3 | 1e-3 | 1e-3 | 1e-4 | 1e-4 | 5e-4 | 1e-3 | 5e-4 |
| Negative Sampling | 512 | 512 | 512 | – | – | 512 | 512 | 512 |
| Hidden Dimension | 1024 | 512 | 512 | 512 | 1024 | 2048 | 2048 | 2048 |
| $\gamma$ | 24.0 | 60.0 | 60.0 | – | – | 9.0 | 9.0 | 9.0 |
| $\lambda_1, \lambda_2, \lambda_3$ | (1.0, 0.4, 0.5) | (0.5, 0.5, 0.5) | (0.5, 0.5, 0.5) | (0.8, 0.6, 0.2) | (0.8, 0.6, 0.2) | (1.0, 0.4, 0.5) | (1.0, 0.4, 0.5) | (1.0, 0.4, 0.5) |
| $\zeta_1, \zeta_2, \zeta_3$ | (0.5, 0.5, 3.5) | (0.5, 0.5, 2.0) | (0.5, 0.5, 2.0) | (0.5, 0.5, 2.0) | (0.5, 0.5, 2.0) | (0.5, 0.5, 4.0) | (0.5, 0.5, 3.5) | (0.5, 0.5, 3.5) |

**YAGO3-10**

|  | TransE | DistMult | ComplEx | ConvE | TuckER | pRotatE | RotatE | HAKE |
|---|---|---|---|---|---|---|---|---|
| Batch Size | 256 | 256 | 256 | 128 | 128 | 256 | 256 | 256 |
| Learning Rate | 1e-3 | 1e-3 | 1e-4 | 5e-5 | 1e-4 | 5e-4 | 1e-3 | 2e-3 |
| Negative Sampling | 256 | 256 | 256 | – | – | 512 | 256 | 512 |
| Hidden Dimension | 1024 | 1024 | 1024 | 512 | 1024 | 1024 | 1024 | 1024 |
| $\gamma$ | 24.0 | 24.0 | 24.0 | – | – | 24.0 | 24.0 | 24.0 |
| $\lambda_1, \lambda_2, \lambda_3$ | (1.0, 0.4, 0.5) | (0.5, 0.5, 0.5) | (1.0, 0.4, 0.5) | (1.0, 0.6, 0.3) | (0.8, 0.6, 0.2) | (1.0, 0.4, 0.5) | (1.0, 0.4, 0.5) | (1.0, 0.4, 0.5) |
| $\zeta_1, \zeta_2, \zeta_3$ | (0.5, 0.5, 2.0) | (0.5, 0.5, 2.0) | (0.2, 0.2, 3.0) | (0.2, 0.5, 2.0) | (0.2, 0.5, 2.0) | (1.0, 1.0, 9.0) | (1.0, 1.0, 9.0) | (1.0, 1.0, 8.0) |

**PrimeKG**

|  | TransE | DistMult | ComplEx | ConvE | TuckER | pRotatE | RotatE | HAKE |
|---|---|---|---|---|---|---|---|---|
| Batch Size | 256 | 512 | 256 | 128 | 128 | 256 | 512 | 512 |
| Learning Rate | 5e-4 | 2e-2 | 2e-3 | 5e-4 | 5e-5 | 1e-4 | 1e-4 | 1e-4 |
| Negative Sampling | 256 | 1024 | 512 | – | – | 256 | 512 | 512 |
| Hidden Dimension | 1024 | 1024 | 2048 | 512 | 1024 | 2048 | 2048 | 2048 |
| $\gamma$ | 24.0 | 200.0 | 200.0 | – | – | 24.0 | 10.0 | 10.0 |
| $\lambda_1, \lambda_2, \lambda_3$ | (1.0, 0.4, 0.5) | (0.8, 0.4, 0.5) | (0.8, 0.4, 0.2) | (1.0, 0.6, 0.2) | (1.0, 0.6, 0.2) | (0.8, 0.8, 0.3) | (0.8, 0.4, 0.2) | (0.8, 0.4, 0.2) |
| $\zeta_1, \zeta_2, \zeta_3$ | (0.5, 0.5, 1.8) | (0.5, 0.5, 2.0) | (0.2, 0.2, 5.0) | (0.5, 0.5, 6.0) | (0.5, 0.5, 6.0) | (0.5, 0.5, 1.8) | (1.0, 1.0, 7.0) | (1.0, 1.0, 9.0) |

## J  Downstream Applications of KG-FIT

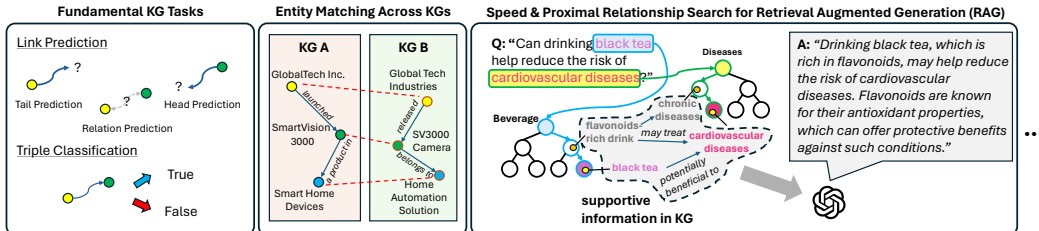

Figure 10: Applications of `KG-FIT`.

The enhanced knowledge graph embeddings produced by `KG-FIT` can enable improved performance on various downstream tasks. As shown in Fig. 10, potential areas of application include:

- **Tasks transformed to fundamental KG tasks.** `KG-FIT`'s strong performance on link prediction can directly benefit fundamental KG tasks like knowledge graph-based question answering (KGQA) [1, 2, 4]. For example, given the question "Can drinking black tea help reduce the risk of cardiovascular diseases?", a KGQA system powered by `KG-FIT` embeddings could effectively traversely perform triple classification task for triples such as *<black tea, potentially beneficial to, cardiovascular diseases>*, *<black tea, is a, flavonoids-rich drink>*, *<flavonoids, may treat, cardiovascular diseases>*, and provide an accurate answer based on the classification results.

- **Entity Matching Across KGs.** `KG-FIT`'s ability to capture both global and local semantics facilitates accurate entity matching across different knowledge graphs [48, 49, 50]. Consider two KGs, A and B, containing information about a company's products. KG A lists the entity "GlobalTech Inc" launched "SmartVision 3000", which is a product in "Smart Home Devices", while KG B mentions "SV3000 Camera" is released by "Global Tech Industries" in the category of "Home Automation Solution". By generating semantically rich embeddings that encode textual, hierarchical and relational information, `KG-FIT` can help identify that "SmartVision 3000" and "SV3000 Camera", "GlobalTech Inc." and "Global Tech Industries" likely refer to the same entities, despite differences in surface form and graph structure.

- **Retrieval Augmented Generation with Graphs.** The hierarchical nature of `KG-FIT`'s embeddings enables efficient search for relevant information to augment language model-based text generation. In a retrieval augmented generation setup, a language model's output can be enhanced by retrieving and conditioning on pertinent information. `KG-FIT`'s embeddings allow for quick identification of relevant entities and relationships via proximity search in the semantic space. Moreover, by traversing the `KG-FIT` hierarchy, the system can gather additional context about an entity of interest. For instance, if the generation task involves the cardiovascular benefits of black tea, searching the `KG-FIT` hierarchy may surface related information on flavonoids and antioxidant properties, providing valuable context to guide the language model in producing an informed and factually grounded response.

- **Other Tasks.** `KG-FIT`'s embeddings can also be leveraged for tasks such as relation extraction [51] and entity disambiguation [52]. By providing high-quality embeddings that encode both local and global information, `KG-FIT` can improve the accuracy and efficiency of these tasks. For example, in relation extraction, `KG-FIT`'s embeddings can help identify the most likely relation between two entities given their positions in the hierarchy and their semantic proximity. In entity disambiguation, `KG-FIT`'s embeddings can be used to disambiguate between multiple entities with the same name by considering their relationships and positions within the knowledge graph hierarchy.

In summary, `KG-FIT`'s robust embeddings, capturing both local and global semantics in a hierarchical structure, can significantly enhance a variety of downstream applications, from fundamental KG tasks to entity matching and retrieval augmented generation. By providing semantically rich and efficiently searchable representations of KG entities and relationships, `KG-FIT` enables knowledge-infused AI systems that can better understand and reason over complex domains.

# K   Interpreting Score Functions in KG-FIT

In this section, we analyze how the entity embeddings in `KG-FIT` are interpreted in the (transitional) score functions defined by different base models.

Let $\mathbf{h}, \mathbf{r}, \mathbf{t} \in \mathbb{R}^{\dim}$ denote the head entity, relation, and tail entity embeddings, respectively. In KG-FIT, the entity embeddings $\mathbf{h}$ and $\mathbf{t}$ are initialized as follows:

$$\mathbf{h} = [\mathbf{h}_n; \mathbf{h}_d], \quad \mathbf{t} = [\mathbf{t}_n; \mathbf{t}_d] \tag{9}$$

where $\mathbf{h}_n, \mathbf{t}_n \in \mathbb{R}^{\dim/2}$ represent the entity name embeddings and $\mathbf{h}_d, \mathbf{t}_d \in \mathbb{R}^{\dim/2}$ represent the entity description embeddings, obtained from the pre-trained text embeddings.

The inclusion of both $\mathbf{h}_n$ and $\mathbf{h}_d$ (similarly for $\mathbf{t}_n$ and $\mathbf{t}_d$) enhances the expressiveness of the entity representation. Starting from these embeddings, KG-FIT effectively captures a more comprehensive understanding of each entity, leading to improved link prediction performance.

**TransE:** The TransE score function is defined as:

$$f_r(\mathbf{h}, \mathbf{t}) = -\|\mathbf{h} + \mathbf{r} - \mathbf{t}\|_p \tag{10}$$

where $\| \cdot \|_p$ denotes the $L_p$ norm.

Expanding the score function using the KG-FIT embeddings:

$$
\begin{aligned}
f_r(\mathbf{h}, \mathbf{t}) &= - \|[\mathbf{h}_n; \mathbf{h}_d] + \mathbf{r} - [\mathbf{t}_n; \mathbf{t}_d]\|_p \\
&= - \|[\mathbf{h}_n + \mathbf{r}_n - \mathbf{t}_n; \mathbf{h}_d + \mathbf{r}_d - \mathbf{t}_d]\|_p
\end{aligned}
\tag{11}
$$

where $\mathbf{r} = [\mathbf{r}_n; \mathbf{r}_d]$ is the relation embedding learned during fine-tuning.

*Interpretation:*

- $\mathbf{h}_n + \mathbf{r}_n - \mathbf{t}_n$: This term represents the distance in the embedding space between the head and tail entities' name embeddings, adjusted by the relation embedding.

- $\mathbf{h}_d + \mathbf{r}_d - \mathbf{t}_d$: This term represents the distance in the embedding space between the head and tail entities' description embeddings, adjusted by the relation embedding.

The TransE score function considers the global semantic information by computing the translation distance between the head and tail entity embeddings, taking into account both the entity name and description embeddings.

**DistMult:** In the DistMult model, the score function is defined as the tri-linear dot product between the head entity embedding $\mathbf{h}$, the relation embedding $\mathbf{r}$, and the tail entity embedding $\mathbf{t}$:

$$f_r(\mathbf{h}, \mathbf{t}) = \langle \mathbf{h}, \mathbf{r}, \mathbf{t} \rangle \tag{12}$$

In KG-FIT, the entity embeddings are initialized by concatenating the entity name embedding and the entity description embedding:

$$\mathbf{h} = [\mathbf{h}_n; \mathbf{h}_d], \quad \mathbf{t} = [\mathbf{t}_n; \mathbf{t}_d] \tag{13}$$

Now, let's expand the DistMult score function by substituting the KG-FIT entity embeddings:

$$
\begin{aligned}
f_r(\mathbf{h}, \mathbf{t}) &= \langle [\mathbf{h}_n; \mathbf{h}_d], \mathbf{r}, [\mathbf{t}_n; \mathbf{t}_d] \rangle \\
&= \langle \mathbf{h}_n, \mathbf{r}_n, \mathbf{t}_n \rangle + \langle \mathbf{h}_d, \mathbf{r}_d, \mathbf{t}_d \rangle \\
&= \sum_{i=1}^{\dim/2} (h_{n,i} \cdot r_{n,i} \cdot t_{n,i}) + \sum_{i=\dim/2+1}^{\dim} (h_{d,i} \cdot r_{d,i} \cdot t_{d,i})
\end{aligned}
\tag{14}
$$

where $\mathbf{r} = [\mathbf{r}_n; \mathbf{r}_d]$ is the relation embedding learned during fine-tuning.

*Interpretation*:

- $\langle \mathbf{h}_n, \mathbf{r}_n, \mathbf{t}_n \rangle$: This term captures the multiplicative interaction between the head entity's name embedding, the relation embedding, and the tail entity's name embedding.

- $\langle \mathbf{h}_d, \mathbf{r}_d, \mathbf{t}_d \rangle$: This term captures the multiplicative interaction between the head entity's description embedding, the relation embedding, and the tail entity's description embedding.

The DistMult score function captures the global semantic information by computing the tri-linear dot product between the head entity, relation, and tail entity embeddings. The dot product considers the interactions between the entity name embeddings and the entity description embeddings separately, allowing the model to capture the global semantic relatedness and attributional similarities.

**ComplEx:** In the ComplEx model, the score function is defined as:

$$
\begin{aligned}
f_r(\mathbf{h}, \mathbf{t}) &= \mathrm{Re}(\langle \mathbf{h}, \mathbf{r}, \overline{\mathbf{t}} \rangle) \\
&= \langle \mathrm{Re}(\mathbf{h}), \mathrm{Re}(\mathbf{r}), \mathrm{Re}(\mathbf{t}) \rangle + \langle \mathrm{Im}(\mathbf{h}), \mathrm{Re}(\mathbf{r}), \mathrm{Im}(\mathbf{t}) \rangle \\
&\quad + \langle \mathrm{Re}(\mathbf{h}), \mathrm{Im}(\mathbf{r}), \mathrm{Im}(\mathbf{t}) \rangle - \langle \mathrm{Im}(\mathbf{h}), \mathrm{Im}(\mathbf{r}), \mathrm{Re}(\mathbf{t}) \rangle
\end{aligned}
\tag{15}
$$

where $\mathrm{Re}(\cdot)$ and $\mathrm{Im}(\cdot)$ denote the real part and imaginary part of a complex number, and $\overline{\mathbf{t}}$ represents the complex conjugate of $\mathbf{t}$.

In KG-FIT, the entity embeddings are initialized as follows:

$$
\mathbf{h} = \mathbf{h}_n + i\mathbf{h}_d, \quad \mathbf{t} = \mathbf{t}_n + i\mathbf{t}_d
\tag{16}
$$

where $\mathbf{h}_n, \mathbf{t}_n \in \mathbb{R}^{\dim/2}$ represent the entity name embeddings (real part) and $\mathbf{h}_d, \mathbf{t}_d \in \mathbb{R}^{\dim/2}$ represent the entity description embeddings (imaginary part).

The relation embedding $\mathbf{r}$ is also a complex-valued vector:

$$
\mathbf{r} = \mathbf{r}_r + i\mathbf{r}_i
\tag{17}
$$

where $\mathbf{r}_r, \mathbf{r}_i \in \mathbb{R}^{\dim/2}$ are learned embeddings that capture the intricate semantics of the relation in the complex space.

Thus, the score function using the KG-FIT embeddings becomes:

$$
\begin{aligned}
f_r(\mathbf{h}, \mathbf{t}) &= \mathrm{Re}(\langle \mathbf{h}, \mathbf{r}, \overline{\mathbf{t}} \rangle) \\
&= \langle \mathbf{h}_n, \mathbf{r}_r, \mathbf{t}_n \rangle + \langle \mathbf{h}_d, \mathbf{r}_r, \mathbf{t}_d \rangle + \langle \mathbf{h}_n, \mathbf{r}_i, \mathbf{t}_d \rangle - \langle \mathbf{h}_d, \mathbf{r}_i, \mathbf{t}_n \rangle \\
&= \mathbf{h}_n \circ \mathbf{r}_r \circ \mathbf{t}_n + \mathbf{h}_d \circ \mathbf{r}_r \circ \mathbf{t}_d + \mathbf{h}_n \circ \mathbf{r}_i \circ \mathbf{t}_d - \mathbf{h}_d \circ \mathbf{r}_i \circ \mathbf{t}_n
\end{aligned}
\tag{18}
$$

*Interpretation:*

- $\mathbf{h}_n \circ \mathbf{r}_r \circ \mathbf{t}_n$ and $\mathbf{h}_d \circ \mathbf{r}_r \circ \mathbf{t}_d$: These terms represent the fundamental interactions between the head and tail entity name embeddings modulated by the real part of the relation. They capture symmetric relationships where the semantic integrity of the relation is maintained irrespective of the direction.

- $\mathbf{h}_n \circ \mathbf{r}_i \circ \mathbf{t}_d$ and $-\mathbf{h}_d \circ \mathbf{r}_i \circ \mathbf{t}_n$: These cross-terms incorporate the imaginary part of the relation embedding, introducing a unique capability to model antisymmetric relations. The inclusion of the imaginary components allows the score function to account for relations where the direction or the orientation between entities significantly alters the meaning or the context of the relation.

**RotatE:** In the RotatE model, each relation is represented as a rotation in the complex plane. The score function is defined as:

$$
\begin{aligned}
f_r(\mathbf{h}, \mathbf{t}) &= -\| \mathbf{h} \circ \mathbf{r} - \mathbf{t} \| \\
&= (\mathrm{Re}(\mathbf{h}) \circ \mathrm{Re}(\mathbf{r}) - \mathrm{Im}(\mathbf{h}) \circ \mathrm{Im}(\mathbf{r}) - \mathrm{Re}(\mathbf{t})) + \\
&\quad + (\mathrm{Re}(\mathbf{h}) \circ \mathrm{Im}(\mathbf{r}) + \mathrm{Im}(\mathbf{h}) \circ \mathrm{Re}(\mathbf{r}) - \mathrm{Im}(\mathbf{t}))
\end{aligned}
\tag{19}
$$

In RotatE with KG-FIT embeddings, we have:

$$
\mathbf{h} = \mathbf{h}_n + i\mathbf{h}_d, \quad \mathbf{t} = \mathbf{t}_n + i\mathbf{t}_d
\tag{20}
$$

and the relation embedding is:

$$
\mathbf{r} = \cos(\theta_r) + i\sin(\theta_r)
\tag{21}
$$

where $\theta_r$ is the learned rotation angle for the relation.

Expanding the RotatE score function using KG-FIT embeddings, we have:

$$
\begin{aligned}
f_r(\mathbf{h}, \mathbf{t}) &= -\|\mathbf{h} \circ \mathbf{r} - \mathbf{t}\| \\
&= -\left[(\mathbf{h}_n \circ \cos(\theta_r) - \mathbf{h}_d \circ \sin(\theta_r) - \mathbf{t}_n) + (\mathbf{h}_n \circ \sin(\theta_r) + \mathbf{h}_d \circ \cos(\theta_r) - \mathbf{t}_d)\right] \\
&= -\left[(\mathbf{h}_n \circ \cos(\theta_r) - \mathbf{t}_n) + (\mathbf{h}_d \circ \cos(\theta_r) - \mathbf{t}_d) + (\mathbf{h}_n \circ \sin(\theta_r)) - (\mathbf{h}_d \circ \sin(\theta_r))\right]
\end{aligned}
$$
(22)

*Interpretation:*

- $\mathbf{h}_n \circ \cos(\theta_r) - \mathbf{t}_n$ and $\mathbf{h}_d \circ \cos(\theta_r) - \mathbf{t}_d$: These terms represent the rotated head entity name and description embeddings respectively, which are then compared with the corresponding tail entity name and description embeddings. The cosine of the relation angle $\theta_r$ scales the embeddings, effectively capturing the strength of the relationship between the entities.

- $\mathbf{h}_n \circ \sin(\theta_r)$ and $-\mathbf{h}_d \circ \sin(\theta_r)$: These terms introduce a phase shift in the entity embeddings based on the relation angle. The sine of the relation angle $\theta_r$ allows for modeling more complex interactions between the head and tail entities, considering both the name and description information.

**pRotatE:** In the pRotatE model, the modulus of the entity embeddings is constrained such that $|h_i| = |t_i| = C$, and the distance function is defined as:

$$
f_r(\mathbf{h}, \mathbf{t}) = -2C \left\| \sin\left(\frac{\theta_h + \theta_r - \theta_t}{2}\right) \right\|_1
$$
(23)

where $\theta_h$, $\theta_r$, and $\theta_t$ represent the phases of the head entity, relation, and tail entity embeddings, respectively. For KG-FIT embeddings, the entity embeddings are complex and represented as:

$$
\mathbf{h} = [\mathbf{h}_n; \mathbf{h}_d], \quad \mathbf{t} = [\mathbf{t}_n; \mathbf{t}_d]
$$
(24)

The phases can be calculated as:

$$
\theta_h = \arg([\mathbf{h}_n; \mathbf{h}_d]), \quad \theta_t = \arg([\mathbf{t}_n; \mathbf{t}_d]), \quad \theta_r = \arg([\mathbf{r}_1; \mathbf{r}_2])
$$
(25)

where $\mathbf{r}_1, \mathbf{r}_2 \in \mathbb{R}^{\dim/2}$ are the learned relation embeddings.

Thus, the pRotatE score function using KG-FIT embeddings becomes:

$$
f_r(\mathbf{h}, \mathbf{t}) = -2C \left\| \sin\left(\frac{\arg([\mathbf{h}_n; \mathbf{h}_d]) + \arg([\mathbf{r}_1; \mathbf{r}_2]) - \arg([\mathbf{t}_n; \mathbf{t}_d])}{2}\right) \right\|_1
$$
(26)

*Interpretation:*

In pRotatE with KG-FIT, the entity phases $\theta_h$ and $\theta_t$ are computed using both the name and description embeddings, while the relation phase $\theta_r$ is learned through the relation embeddings $\mathbf{r}_1$ and $\mathbf{r}_2$.

The model aims to minimize the phase difference $\left\| \sin\left(\frac{\theta_h + \theta_r - \theta_t}{2}\right) \right\|_1$ for valid triples, considering both the entity name and description information. This allows pRotatE to capture the complex interactions between entities and relations in the knowledge graph.

**HAKE:** In the HAKE (Hierarchy-Aware Knowledge Graph Embedding) model, entities are embedded into polar coordinate space to capture hierarchical structures. The score function is a combination of radial and angular distances:

$$
f_r(\mathbf{h}, \mathbf{t}) = -\alpha \|\mathbf{h}_{\mathrm{mod}} \circ \mathbf{r}_{\mathrm{mod}} - \mathbf{t}_{\mathrm{mod}}\|_2 - \beta \left\| \sin\left(\frac{\mathbf{h}_{\mathrm{phase}} + \mathbf{r}_{\mathrm{phase}} - \mathbf{t}_{\mathrm{phase}}}{2}\right) \right\|_1
$$
(27)

In KG-FIT, $\mathbf{h}_{\mathrm{mod}} = \mathbf{h}_d$, $\mathbf{t}_{\mathrm{mod}} = \mathbf{t}_d$, $\mathbf{h}_{\mathrm{phase}} = \arg(\mathbf{h}_n)$, and $\mathbf{t}_{\mathrm{phase}} = \arg(\mathbf{t}_n)$. The learned relation embedding is $\mathbf{r} = [\mathbf{r}_n; \mathbf{r}_d]$.

Thus, the HAKE score function using KG-FIT embeddings becomes:

$$
f_r(\mathbf{h}, \mathbf{t}) = -\alpha \|\mathbf{h}_d \circ \mathbf{r}_d - \mathbf{t}_d\|_2 - \beta \left\| \sin\left(\frac{\arg(\mathbf{h}_n) + \arg(\mathbf{r}_n) - \arg(\mathbf{t}_n)}{2}\right) \right\|_1
$$
(28)

*Interpretation:*

In HAKE with KG-FIT, the entity description embeddings are used to determine the modulus, while the entity name embeddings are used to determine the phase. This approach seamlessly utilizes the information from both types of embeddings from pre-trained language models.

# L Notation Table

Table 14 provides a comprehensive list of the notations used throughout this paper, along with their corresponding descriptions. This table serves as a quick reference to help readers better understand the concepts presented in our work.

Table 14: Notations and Descriptions in KG-FIT

| Notation | Description |
|---|---|
| $\mathcal{E}$ | Set of entities in the knowledge graph |
| $e_i \mid d_i$ | The $i$-th entity in the knowledge graph \| Text description of entity $e_i$ |
| $f$ | Text embedding model |
| $\mathbf{v}_i^e \mid \mathbf{v}_i^d$ | Text embedding of entity $e_i$ \| Text embedding of description $d_i$ |
| $\mathbf{v}_i$ | Concatenated embedding of entity $e_i$ and description $d_i$ |
| $\tau$ | Clustering threshold |
| $\tau_{\min}, \tau_{\max} \mid \tau_{\text{optim}}$ | Minimum and maximum clustering thresholds \| Optimal clustering threshold |
| $S^*$ | Silhouette score |
| $\text{labels}_\tau$ | Clustering results at distance threshold $\tau$ |
| $\mathcal{H}_{\text{seed}}$ | Seed hierarchy constructed by agglomerative clustering |
| $C_{\text{original}} \mid C_{\text{split}}$ | Original cluster in the seed hierarchy \| New clusters after splitting $C_{\text{old}}$ |
| $\mathcal{C}_{\text{leaf}}$ | Set of leaf clusters in the seed hierarchy |
| $C_i$ | The $i$-th new cluster after splitting |
| $k$ | Number of new clusters after splitting |
| $\mathcal{H}_{\text{split}}$ | Intermediate hierarchy after cluster splitting |
| $\mathbb{T}_{\mathcal{H}_{\text{split}}}$ | Set of parent-child triples in the intermediate hierarchy |
| $(P_*, P_l, P_r)$ | Parent-child triple (sub-tree) in the hierarchy |
| $\mathcal{H}_{\text{LHR}}$ | LLM-guided refined hierarchy |
| $\mathbf{v}_i'$ | Sliced text embedding of entity $e_i$ |
| $\mathbf{e}_i'$ | Randomly initialized embedding of entity $e_i$ |
| $\rho$ | Hyperparameter controlling the density of randomized embedding |
| $\mathbf{e}_i$ | KG-FIT embedding of entity $e_i$ |
| $\mathbf{r}_j$ | Embedding of relation $j$ |
| $n \mid m$ | Dimension of entity embedding $\mathbf{e}_i$ \| Dimension of relation embedding $\mathbf{r}_j$ |
| $\psi$ | Hyperparameter controlling the standard deviation |
| $C \mid C'$ | Cluster that an entity belongs to \| Neighbor cluster of $C$ |
| $\mathbf{c} \mid \mathbf{c}'$ | Cluster embedding of $C$ \| Cluster embedding of $C'$ |
| $\mathcal{S}_m(C)$ | Set of $m$ nearest neighbor clusters of $C$ |
| $P$ | Parent nodes |
| $\mathbf{p}_j, \mathbf{p}_{j+1}$ | Embeddings of successive parent nodes along the path from an entity to the root |
| $d(\cdot, \cdot)$ | Distance function for measuring distances between embeddings |
| $\beta_j$ | Weight for the distance to the $j$-th parent node |
| $\beta_0$ | Initial weight for the closest parent node |
| $\phi$ | Decay rate for hierarchical distance maintenance |
| $\lambda_1, \lambda_2, \lambda_3$ | Hyperparameters for the hierarchical clustering constraint |
| $\mathcal{D}$ | Set of all triples in the knowledge graph |
| $\sigma$ | Sigmoid function |
| $\gamma$ | Margin hyperparameter for link prediction |
| $f_r(\cdot, \cdot)$ | Scoring function defined by structure-based models |
| $\mathcal{N}_j$ | Set of negative tail entities sampled for a triple $(e_i, r, e_j)$ |
| $n_j$ | Negative tail entity |
| $\mathbf{e}_{n_j}$ | Embedding of the negative tail entity $n_j$ |
| $\zeta_1, \zeta_2, \zeta_3$ | Hyperparameters for weighting the training objectives |
| $L$ | Average sequence length in PLM-based methods |
| $n_{\text{PLM}}$ | Hidden dimension of the PLM in PLM-based methods |

