# OpenReview forum: "KG-FIT: Knowledge Graph Fine-Tuning Upon Open-World Knowledge"
_NeurIPS.cc/2024/Conference — NeurIPS 2024 poster_

### Official Review · Reviewer_9oop · 2024-07-08

**Soundness:** 3
**Presentation:** 4
**Contribution:** 3
**Rating:** 6
**Confidence:** 4

**Summary:**

This paper introduces KG-FIT, a general framework that enhances the expressiveness of existing Knowledge Graph Embedding (KGE) models by integrating LLMs. KG-FIT contains four key steps: First, it utilizes an LLM to generate descriptions for a set of given entities, forming an enriched entity representation by concatenating the entity's embedding with its description. Second, it constructs a semantically coherent seed hierarchical structure. Third, it leverages the real-world entity knowledge captured by the LLM to refine this hierarchical structure. Finally, it fine-tunes the knowledge graph embeddings by integrating the hierarchical structure with textual embeddings. Extensive experiments validate the effectiveness of KG-FIT.

**Strengths:**

1. The motivation of the proposed KG-FIT is clear and the paper is well-structured.
2. Extensive experimental results demonstrate that KG-FIT can improve most KGE baseline models.
3. Codes are provided for reproducibility.

**Weaknesses:**

1. The performance of KG-FIT heavily relies on the LLM used for generating entity descriptions and guiding the refinement of the seed hierarchy refinement. If the LLM lacks comprehensive real-world entity knowledge about the given entities or domains, the resulting embeddings may not suboptimal. Furthermore, this dependency could make KG-FIT less effective in the situations where LLM have limited coverage.
2. If the LLM has limited coverage of a specific domain, the LLM guided hierarchy refinement process may yield incorrect results, potentially distorting rather than enhancing the structure of the well-formed seed hierarchy.
3. The paper does not justify the selection of agglomerative hierarchical clustering and the use of silhouette score. Additional ablation studies will enhance this work.

**Questions:**

1. It is unclear the explicit description of $P$, $L$, $R$ in line 134.
2. It remains unclear whether the efficiency evaluation of training time contains the duration required for seed hierarchy construction and LLM-guided hierarchy refinement stages.

**Limitations:**

The authors have discussed the limitations of their work in the paper.

---

> ### Author Rebuttal · Authors · 2024-08-06
>
> We appreciate the reviewer's thoughtful comments and their recognition of our work's strengths. We address their concerns as follows:
>
> ---
>
> > ### **[W1/W2]** (Dependency on LLM knowledge and potential issues with limited domain coverage)
>
> We appreciate the reviewer's concern about KG-FIT's reliance on LLM knowledge. While we view this as a strength that allows our method to leverage vast and continually improving knowledge, we acknowledge the potential limitations in highly specialized domains.
>
> To address this, we propose several mitigating strategies:
>
> 1. Incorporating external knowledge: We can enhance LLM performance in specialized domains by using Retrieval-Augmented Generation (RAG) to incorporate domain-specific external knowledge bases during the description generation process.
>
> 2. Leveraging KG context: For domains where external knowledge is limited, we can use the context within the Knowledge Graph itself to generate more informative entity descriptions. This approach ensures that even without extensive domain knowledge, the LLM can still provide useful descriptions based on the relationships and attributes present in the KG.
>
> 3. Fallback to seed hierarchy: In cases where the LLM truly lacks domain-specific knowledge, our results show that the seed hierarchy alone still significantly improves KG embeddings. As demonstrated in Table 2 of our paper, KG-FIT with just the seed hierarchy (before LLM refinement) consistently outperforms base models across all datasets.
>
> 4. Domain-specific LLMs: When available, using domain-specific LLMs can provide more accurate and relevant knowledge for specialized fields.
>
> These strategies ensure that KG-FIT remains effective and adaptable across a wide range of domains, from general knowledge to highly specialized fields. Our experiments on diverse datasets (FB15K-237, YAGO3-10, PrimeKG) demonstrate KG-FIT's robustness and broad applicability, even when dealing with domain-specific knowledge.
>
> In future work, we plan to explore methods for automatically selecting the most appropriate strategy based on the domain and available resources, further enhancing KG-FIT's versatility and performance.
>
> > ### **[W3]** (Justification for agglomerative clustering and silhouette score)
>
> We appreciate the reviewer's suggestion for additional justification. We chose agglomerative clustering for its natural ability to create a hierarchical structure without pre-specifying the number of clusters, which is crucial for our approach. The silhouette score was selected as it balances both cluster cohesion and separation, providing a robust measure of clustering quality.
>
> To address the reviewer's concern, we will conduct additional ablation studies comparing different clustering methods:
>
> 1. Top-down agglomerative clustering: This variant will start with all entities in one cluster and progressively split them, potentially offering a different perspective on the hierarchy.
> 2. K-means: We will implement a recursive K-means process, where we first cluster all entities, then recursively apply K-means to each resulting cluster until a stopping criterion is met (e.g., cluster size or maximum depth). This will create a top-down hierarchical structure.
> 3. DBSCAN: We will use a similar recursive approach as with K-means, but DBSCAN's ability to detect noise points will allow us to create a hierarchy that potentially captures outliers at higher levels.
>
> For evaluation metrics, alongside the silhouette score, we will also compare the Calinski-Harabasz index and Davies-Bouldin index.
>
> We will include these results in the appendix of our revised paper, providing a comprehensive comparison of different clustering methods and their impact on KG-FIT's performance.
>
> ---
>
> > ### **[Q1]** (Clarification on P, L, R in line 134)
>
> We apologize for the lack of clarity. P, L, and R refer to Parent, Left child, and Right child, respectively. We will add this explanation to the paper for better understanding.
>
> > ### **[Q2]** (Clarification on efficiency evaluation)
>
> The reported training time in Table 4 indeed focuses on the fine-tuning stage for a fair comparison with baselines. The hierarchy construction (Steps 1-3) is a one-time preprocessing step. For transparency, we will add the following breakdown to Appendix H:
>
> - Entity description & text embedding generation (with 15 threads): ~10 minutes for FB15K-237, ~1 hour for YAGO3-10, ~8 minutes for PrimeKG
> - Seed hierarchy construction: ~2 minutes for FB15K-237, ~8 minutes for YAGO3-10, ~1.5 minutes for PrimeKG
> - LLM-guided refinement (with 15 threads): ~12 minutes for FB15K-237,  ~1 hour for YAGO3-10, ~10 minutes for PrimeKG
>
> These preprocessing times are relatively small compared to the overall training process, especially considering they're one-time operations that can be reused for multiple experiments or model variations. Moreover, the LLM-guided refinement step shows good scalability with parallel processing, which can further reduce preprocessing time for larger datasets.

---

> > ### Comment · Reviewer_9oop · 2024-08-09
> > **Thanks for the response**
> >
> > Thanks for the response. I have read all the reviews and responses, and am satisfied with the response to my comments. I will maintain my positive review score.

---

> > > ### Author Response · Authors · 2024-08-09
> > > **Thank You for Your Recognition**
> > >
> > > Thank you for your thoughtful review and for recognizing the strengths of our work! We appreciate your positive evaluation and are glad our responses addressed your concerns. Your feedback has been invaluable in refining our approach. Please let us know whenever you have any further questions during this reviewer-author discussion period. We are happy to discuss and provide any additional information.
> > >
> > > Thank you again for your support, and we look forward to continuing our research in this exciting area.

---

### Official Review · Reviewer_dKZ2 · 2024-07-10

**Soundness:** 3
**Presentation:** 2
**Contribution:** 2
**Rating:** 4
**Confidence:** 5

**Summary:**

This paper addresses the limitations of existing KGE models that focus either on graph structure or fine-tuning pre-trained language models. It introduces KG-FIT, which leverages LLM-guided refinement to incorporate hierarchical and textual knowledge, effectively capturing both global and local semantics. Experiments on benchmark datasets demonstrate KG-FIT's superiority, achieving significant performance improvements over state-of-the-art methods.

**Strengths:**

1. The proposed method can automatically construct a semantically coherent entity hierarchy using agglomerative clustering and LLM-guided refinement, which is an interesting topic.
2. The authors provide detailed illustrations for extensive empirical study on benchmark datasets and demonstrate significant improvements in link prediction accuracy.

**Weaknesses:**

1. The paper is not organized clearly, which is not friendly for understanding. For example, there is a lack the sensitivity study for the hyperparameters in the loss function.
2. The comparable methods are old and lack the new ones in the last 2 years such as [1][2][3][4]. The performance is not comparable with the previous work.
[1] Compounding Geometric Operations for Knowledge Graph Completion
[2] Geometry Interaction Knowledge Graph Embeddings
[3] KRACL: Contrastive Learning with Graph Context Modeling for Sparse Knowledge Graph Completion
[4] Dual Quaternion Knowledge Graph Embeddings
3. The paper lacks the analysis of time complexity as well as space complexity, which is necessary to study the efficiency of the model.
4. There are some typos and It is commanded that the writing needs to be improved.
(1) On page 5, line 163 “determined by lowest common ancestor” should be “determined by the lowest common ancestor”
(2) On page 6, line 195 “is sigmoid function” should be “is the sigmoid function”

**Questions:**

Please refer to the weaknesses.

---

> ### Author Rebuttal · Authors · 2024-08-05
>
> Thank you for your thoughtful review. We appreciate your recognition of our work's strengths and we address your concerns as follows.
>
> ---
>
> > ### **[W1]** *"there is a lack of the sensitivity study for the hyperparameters in the loss function"*
>
> We appreciate the reviewer's concern about the lack of a sensitivity study. To address this, we conducted a sensitivity analysis, presented in **Figure A in our rebuttal PDF**. We have also presented our hyperparameter study in **Appendix I in our paper**.
>
> Figure A demonstrates KG-FIT's robustness to hyperparameter variations:
>
> 1. Left plot (λ1, λ2, λ3): Performance metrics remain stable across different combinations, indicating that KG-FIT is not overly sensitive to these hierarchical clustering constraint parameters.
>
> 2. Right plot (ζ1, ζ2, ζ3): While there's some variation, performance remains consistently high over a range of ratios for these loss function component weights.
>
> These results show that KG-FIT maintains good performance across various hyperparameter settings, addressing the reviewer's concern and suggesting that the model can be readily applied to new datasets without requiring extensive tuning.
>
> > ### **[W2]** *"The comparable methods are old and lack the new ones in the last 2 years"*
>
> Thank you for suggesting these recent baselines. We have conducted additional experiments to compare KG-FIT with CompoundE [1], GIE [2], and DualE [4]. The results are presented in **Table C in our rebuttal PDF**.
>
> These results demonstrate that KG-FIT not only compares favorably with but substantially improves upon recent state-of-the-art KGE methods. This underscores KG-FIT's effectiveness in leveraging LLM knowledge to enhance various KGE architectures.
>
> Regarding KRACL [3], as it employs a GNN-based approach, integrating it with KG-FIT presents unique challenges. We have added the exploration of KG-FIT's integration with GNN-based methods to our future work list.
>
> We appreciate the reviewer's suggestion to include these recent baselines, as it has allowed us to further demonstrate KG-FIT's capabilities and versatility.
>
>
>
> > ### **[W3]** *"The paper lacks the analysis of time complexity as well as space complexity"*
>
> We appreciate the reviewer's attention to this important aspect of our model. Here's a clarification:
>
> **For time complexity:**
> We have analyzed KG-FIT's time complexity both theoretically (Lines 206-210) and empirically (Table 4, Lines 296-303). Our results demonstrate that KG-FIT is 12 times faster in training than the best PLM-based method, while maintaining inference speed comparable to backbone KGE methods.
>
> **For space complexity:**
> While we didn't explicitly state it in the paper, the space complexity of KG-FIT in terms of trainable parameters is the same as the backbone KGE models:
>
> $O(|E| * n + |R| * m)$
>
> Where $|E|$ is the number of entities, $|R|$ is the number of relations, $n$ is the entity embedding dimension, and $m$ is the relation embedding dimension.
>
> This is because the main trainable components of KG-FIT are:
>
> 1. Entity embeddings: $O(|E| * n)$
> 2. Relation embeddings: $O(|R| * m)$
>
> The additional components introduced by KG-FIT (entity text embeddings and cluster embeddings) are not trainable parameters, but fixed inputs used during the forward pass. They do consume memory during runtime but do not increase the model's parameter count.
>
> This space complexity is significantly lower than PLM-based methods, which often require gigabytes of memory for model parameters alone. For example, on the FB15K-237 dataset, KG-FIT's trainable parameters would only occupy approximately 60MB of memory (assuming 32-bit floating-point numbers and $n = m = 1024$).
>
> > ### **[W4]** (Typos in the paper)
>
> We appreciate the reviewer's attention to detail. We will correct these typos:
>
> 1. Page 5, line 163: "determined by the lowest common ancestor"
> 2. Page 6, line 195: "is the sigmoid function"
>
> We will thoroughly proofread the entire manuscript to improve clarity and precision. Thank you for helping us enhance the quality of our paper.
>
>
>
>
>
> ---
>
> **References**
>
> [1] (ACL 2023) Compounding Geometric Operations for Knowledge Graph Completion.
>
> [2] (AAAI 2022) Geometry Interaction Knowledge Graph Embeddings.
>
> [3] (WWW 2023) KRACL: Contrastive Learning with Graph Context Modeling for Sparse Knowledge Graph Completion.
>
> [4] (AAAI 2021) Dual Quaternion Knowledge Graph Embeddings.

---

> ### Author Response · Authors · 2024-08-13
>
> Dear Reviewer dKZ2,
>
> Thank you for your insightful comments and suggestions. In our author response, we have provided additional experimental results, analyses, and clarifications to address your concerns.
>
> As the **discussion period nears its end (in 24 hours)**, we would be grateful if you could take a moment to review our response and let us know if there are any remaining concerns or if our clarifications have adequately addressed your points.
>
> We are grateful for the time and expertise you have shared in reviewing our work.
>
> Sincerely,
>
> The Authors

---

### Official Review · Reviewer_G3dr · 2024-07-12

**Soundness:** 3
**Presentation:** 3
**Contribution:** 2
**Rating:** 5
**Confidence:** 4

**Summary:**

Knowledge graphs (KGs) are essential for representing structured knowledge in various domains. They consist of entities and relations, forming a graph structure for efficient reasoning and knowledge discovery. Current knowledge graph embedding (KGE) methods create low-dimensional representations of these entities and relations but often overlook extensive open-world knowledge, limiting their performance. Pre-trained language models (PLMs) and large language models (LLMs) offer a broader understanding but are computationally expensive to fine-tune with KGs. To address these issues, the authors propose KG-FIT, a framework that incorporates rich knowledge from LLMs into KG embeddings without fine-tuning the LLMs. KG-FIT generates entity descriptions from LLMs, constructs a hierarchical structure of entities, and fine-tunes KG embeddings by integrating this hierarchy with textual embeddings. This approach enhances KG representations, combining global knowledge from LLMs with local KG knowledge, significantly improving link prediction accuracy on benchmark datasets.

**Strengths:**

1. Extensive experiments. The authors compare experiment performance with 8 baselines on three datasets and apply on 8 KG embedding backbones.


2. Extensive experimental details description. For example, the hardware environment for running the experiment, data processing, prompts for interacting with large models, and codes to reproduce their results.

3. Clear figures and presentations.

**Weaknesses:**

1.  The motivation needs to be reconsidered. The authors mention that using KG to fine-tune LLMs is computationally expensive. Many current research efforts do not fine-tune LLMs with KG. Instead, they use retrieval-based methods to explicitly provide the knowledge.

2. LLM-based baselines should be considered. The authors extensively use LLMs in their methods. They also should incorporate the LLM-based methods. For example, those LLMs retrieval-based methods. Only comparing with small LM-based methods is not enough.

3. The contribution is kind of limited. I do not know why the authors still use KG embedding methods for efficient reasoning and knowledge discovery since LLMs have very strong reasoning ability for knowledge discovery.

**Questions:**

1. Could the authors reconsider the motivation behind their approach?

2. Why haven’t the authors considered incorporating published LLM-based baselines?

3. Why do the authors continue to use KG embedding methods for efficient reasoning and knowledge discovery when LLMs possess very strong reasoning abilities for knowledge discovery?

**Limitations:**

Please refer to my weakness

---

> ### Author Rebuttal · Authors · 2024-08-04
>
> Thank you for your thoughtful review. We appreciate your recognition of our work's strengths and we address your concerns as follows.
>
> ---
>
> > ### **[W1/Q1]** (Reconsidering motivation)
>
> If the retrieval process here refers to the "retriever" mentioned in KICGPT [1], it's important to note that:
>
> - The retriever in [1] is actually a KG embedding (KGE) model (RotatE in their case) used to generate top-ranked candidate object entities $o$ given a query $(s, r, ?)$. The LLM is then used to re-rank these candidates.
>
> - In this case, the final results rely heavily on the candidate generation handled by the KGE methods. This means that re-ranking methods like [1, 2, 3] can be used as an add-on to KG-FIT, where KG-FIT provides more accurate candidates, improving the final results. We demonstrate this by implementing KICGPT with KG-FIT and show the results in **Table B of our rebuttal PDF**.
>
> However, if the retrieval process does not involve KGE methods:
>
> 1. Link prediction is a crucial task for evaluating knowledge discovery ability. To the best of our knowledge, there are currently no methods that use LLM with retrieval for this task.
> 2. Knowledge discovery on an existing KG typically requires the model to be fine-tuned on the KG. This is because (1) we need a comprehensive view of all possible object entities $o$ given a query $(s, r, ?)$, and (2) the model must learn underlying patterns from the existing knowledge.
> 3. Traditional retrieval methods without KGE cannot be directly applied for link prediction because they do not provide a systematic way to rank all possible object entities for a given query. They typically retrieve a small subset of relevant entities based on text similarity, which may miss many valid candidates. In contrast, KGE methods like KG-FIT learn to embed the entire KG structure, enabling them to score and rank all possible object entities for a given query.
>
> We will clarify these points in the revised paper to highlight the unique strengths of KG-FIT and its potential to complement retrieval-based methods.
>
> > ### **[W2/Q2]** (Incorporating LLM-based baselines)
>
> Thank you for this suggestion. We have added two recent LLM-based baselines: KICGPT [1] and KG-LLM [4].
>
> The results are shown in **Table B of our rebuttal PDF** and will be added to Table 2 in the revised paper. Note that we only implemented KG-LLM on FB15K-237 due to its high computational cost.
>
> > ### **[W3/Q3]** (Why KG embedding methods)
>
> We appreciate this question as it allows us to clarify the unique advantages of our approach. Let's compare different methods:
>
> 1. Fine-tuning-based methods:
>    - Pros:
>      - Can adapt LLMs to specific KG domains
>      - Leverage the vast knowledge and reasoning capabilities of LLMs
>    - Cons:
>      - Computationally expensive, especially for large LLMs
>      - May overfit to small KGs due to the large number of parameters
>      - Difficult to update as the KG evolves, requiring retraining
> 2. Re-ranking (Retrieval)-based methods:
>    - Pros:
>      - Leverage LLM knowledge without expensive fine-tuning
>      - Computationally efficient for inference
>    - Cons:
>      - Rely on pre-existing KG embeddings for candidate generation
>      - May miss global patterns and relationships in the KG
>      - Limited by the quality of the retrieval process and the retrieved context
> 3. KG-FIT and KG embedding methods:
>    - Pros:
>      - Capture the global structure and patterns of the entire KG
>      - Computationally efficient
>      - Easily updatable as new knowledge is added to the KG
>      - Provide interpretable entity and relation representations
>    - Cons:
>      - Require a well-designed model architecture and training process to effectively incorporate LLM knowledge
>
> To illustrate the importance of KGE methods for knowledge discovery, let's consider the following example.
>
> > **[Example]** Suppose we have a movie knowledge graph where entities represent actors, movies, directors, and genres, and relations represent facts like "acted_in", "directed_by", and "belongs_to_genre". Given a query $(s, r, ?)$ where $s$ is a specific actor, $r$ is "likely_to_collaborate_with", and the goal is to predict another actor $o$ that $s$ is likely to work with in the future, a KGE model like KG-FIT can learn from the global KG structure to infer patterns such as "actors who have worked with the same directors or in similar genres are more likely to collaborate". This allows KG-FIT to make informed predictions about potential actor collaborations, even if those specific actors have never worked together before. In contrast, a non-fine-tuned LLM-based method might have difficulty making such inferences if the actors' previous collaborations are not explicitly mentioned in the training set/retrieved context, as it lacks the global understanding of the interconnected nature of the film industry that KGE methods can capture.
>
> It is also important to note that **KG embeddings enhanced by KG-FIT are complementary to LLM/PLM fine-tuning and re-ranking methods**. For example, PKGC [2] and TagReal [3] apply both fine-tuning and re-ranking, but they heavily rely on the "retrieved" results from their backbone KG embeddings. KG-FIT can  improve the quality of these backbone embeddings, leading to better overall performance.
>
> Moreover, KG-FIT strikes a balance between leveraging the strengths of LLMs and maintaining the desirable properties of KG embeddings. By incorporating LLM knowledge into the embedding space, KG-FIT can capture more semantic information and complex relationships while still being computationally efficient and interpretable.
>
> ---
>
> **References**
>
> [1] (EMNLP 2023) KICGPT: Large language model with knowledge in context for knowledge graph completion.
>
> [2] (ACL 2022) Do pre-trained models benefit knowledge graph completion?
>
> [3] (ACL 2023) Text-augmented open knowledge graph completion via pre-trained language models.
>
> [4] (arXiv) Exploring large language models for knowledge graph completion.

---

> > ### Comment · Reviewer_G3dr · 2024-08-11
> >
> > Thanks for the response. After carefully considering your feedback as well as the comments from other reviewers, I have decided to maintain my rating since this paper does not yet meet the rigorous standards expected for publication in NeurIPS. I'd like to see an improved version after a major revision.

---

### Official Review · Reviewer_hiNn · 2024-07-15

**Soundness:** 2
**Presentation:** 3
**Contribution:** 2
**Rating:** 5
**Confidence:** 4

**Summary:**

The paper introduces a framework called KG-FIT for enhancing knowledge graph embeddings by integrating knowledge from large language models (LLMs). KG-FIT enriches entity descriptions using LLMs and then constructs a semantically coherent hierarchical structure of entities. It finally fine-tunes KG embeddings using this hierarchy and textual information. Experiments on benchmark datasets (FB15K-237, YAGO3-10, PrimeKG) demonstrate its effectiveness in link prediction.

**Strengths:**

- Using LLMs to enhance KG embeddings can capture comprehensive features. The proposed method demonstrates improvements in link prediction when compared to selected baseline methods.

- The presentation of the paper is good.

**Weaknesses:**

- There are several related studies on using LLMs to enhance text information in KGs [1,2], which, however, were not discussed in the paper. Additionally, constructing a hierarchy seems redundant given the existing graph structure of the KG. More discussion and explanation are needed.

- The rationale behind the technical design is unclear. For instance, the proposed method concatenates structural and textual embeddings to construct the hierarchy, and then linearly combines these embeddings for fine-tuning. What are the underlying reasons for these choices?

- The method may be computationally expensive due to the use of LLMs and hierarchical refinement. In my view, it is not profitable to use LLMs to achieve ~0.02 Hits@1 improvements in link prediction.

- Several recent strong baselines for KG link prediction, such as NBFNet [3] and AdaProp [4], which both achieve over 0.32 Hits@1 on FB15K-237, are absent from the experiments. It remains uncertain whether the proposed method can still improve the two baselines.

[1] Derong Xu, Ziheng Zhang, Zhenxi Lin, Xian Wu, Zhihong Zhu, Tong Xu, Xiangyu Zhao, Yefeng Zheng, Enhong Chen: Multi-perspective Improvement of Knowledge Graph Completion with Large Language Models. LREC/COLING 2024: 11956-11968

[2] Dawei Li, Zhen Tan, Tianlong Chen, Huan Liu: Contextualization Distillation from Large Language Model for Knowledge Graph Completion. EACL (Findings) 2024: 458-477

[3] Zhaocheng Zhu, Zuobai Zhang, Louis-Pascal A. C. Xhonneux, Jian Tang: Neural Bellman-Ford Networks: A General Graph Neural Network Framework for Link Prediction. NeurIPS 2021: 29476-29490

[4] Yongqi Zhang, Zhanke Zhou, Quanming Yao, Xiaowen Chu, Bo Han: AdaProp: Learning Adaptive Propagation for Graph Neural Network based Knowledge Graph Reasoning. KDD 2023: 3446-3457

**Questions:**

- The proposed method leverages entity names to prompt large language models (LLMs) to generate corresponding descriptions. How does it address the issue of multiple entities having the same name?

- Please see other questions in "Weaknesses".

**Limitations:**

NA.

---

> ### Author Rebuttal · Authors · 2024-08-02
>
> Thank you for your thoughtful review. We appreciate your recognition of our work's strengths and we address your concerns as follows.
>
> ---
>
> > ### **[W1.1]** *"There are several related studies on using LLMs to enhance text information in KGs [1,2] ..."*
>
> We acknowledge this oversight. In our latest draft, we have cited [1] and [2] in line 83:
>
> - "Some methods [1,2] are proposed to improve the performance of the aforementioned methods by enhancing the text information of entities/relations in KGs."
>
> Interestingly, these methods are complementary to Step 1 of KG-FIT. We conducted additional experiments replacing our entity descriptions with those generated by MPIKGC [1] (E&S strategy) and Contextualization Distillation (CD) [2] (ED strategy). For CD, we averaged the embeddings of all descriptions generated for each entity.
>
> Results on FB15K-237 with RotatE and HAKE are shown in **Table A in our rebuttal PDF**, which demonstrates that KG-FIT can be further enhanced by incorporating these methods.
>
>
> > ### **[W1.2]** *"Additionally, constructing a hierarchy seems redundant given the existing graph structure of the KG."*
>
> Constructing a hierarchy offers key benefits:
>
> 1. **Different levels of abstraction:**
>    KGs represent direct relationships but not hierarchical relationships or varying abstraction levels. KG-FIT's hierarchy captures broader semantic relationships not directly encoded in the KG.
>    - *Example:* In a biomedical KG, "Aspirin" and "Ibuprofen" might be directly connected as "pain relievers". However, an LLM-constructed hierarchy could group them under "NSAIDs", then "Analgesics", and finally "Pharmaceuticals", providing a richer semantic context.
> 2. **Incorporating external knowledge:**
>    The LLM-constructed hierarchy in KG-FIT incorporates external knowledge absent from the original KG, enriching entity representations.
>    - *Example:* "Apple" and "Microsoft" grouped under "Tech Giants", "Consumer Electronics", and "Fortune 500 Companies", incorporating broader market knowledge not explicit in the original KG.
> 3. **Handling sparse connections and improved generalization:**
>    KGs often have sparse entity connections. Hierarchical structure bridges gaps between semantically related but unconnected entities and enhances generalization to unseen entities or relationships.
>    - *Example:* In a medical KG, "Heart Disease" and "Diabetes" might not be directly connected, but both could be grouped under "Chronic Diseases". An LLM-constructed hierarchy could further classify them under "Cardiovascular Diseases" and "Metabolic Disorders", respectively. This organization bridges gaps and generalizes treatment or risk factors shared among similar diseases, providing a richer context for inference.
>
> These hierarchies provide valuable semantic context beyond the explicit KG structure, enhancing overall embedding representational power.
>
> We have also analyzed the effect of hierarchy in **Table 3 and Figure 3 in the paper**.
>
>
>
> > ### **[W2]** *"The proposed method concatenates structural and textual embeddings to construct the hierarchy, ..."*
>
> This is a misunderstanding. Our method involves two distinct steps:
>
> 1. **Hierarchy construction**: We use **only textual information** - *no structural embeddings*. The enriched entity embedding $v_i$ is a **concatenation of entity name embedding $v^e_i$ and description embedding $v^d_i$** (Equation 1).
>
> 2. **Fine-tuning**: Entity embedding $e_i$ is **initialized as a linear combination** of a random embedding $e'_i$ and sliced text embedding $v'_i$ (Equation 5). This allows the integration of *LLM-derived semantics while adapting to KG structure*.
>
> Structural information from the KG is used *only in the link prediction objective (Equation 8)*.
>
>
>
> > ### **[W3]** *"Expensive use of LLMs and hierarchical refinement."*
>
> We appreciate this concern, but we respectfully disagree for two main reasons:
>
> 1. **Significant improvements**: In many cases the improvements are substantial. For example, KG-FIT-HAKE w/ LHR on PrimeKG achieves a ~0.07 improvement in Hits@1 over KG-FIT-HAKE w/ seed hierarchy.
> 2. **One-time, reasonable cost**: The use of LLMs is limited to the hierarchy construction, which is a one-time preprocessing step. As detailed in Appendix H.2, even for large-scale KGs like YAGO3-10, the cost remains reasonable when considering the deployment of KG-FIT in real-world applications.
>
>
>
> > ### **[W4]** *"Several recent strong baselines for KG link prediction, such as NBFNet [3] and AdaProp [4] ..."*
>
> We appreciate the suggestion to compare with these strong GNN-based baselines. However, KG-FIT is designed for embedding-based methods, offering better scalability, interpretability, and lower computational requirements. Adapting KG-FIT to NBFNet or AdaProp would require substantial changes and might lose these benefits.
>
> KG-FIT constructs a hierarchical structure of entity clusters using **static embeddings,** while NBFNet and AdaProp use **dynamic entity representations** through message passing. Integrating these fundamentally different architectures would essentially amount to developing a new hybrid model, beyond our current scope.
>
> Nonetheless, we believe KG-FIT's underlying philosophy could inspire improvements in GNN-based methods in future work.
>
> ---
>
> > ### **[Q1]** *"Issue of multiple entities having the same name"*
>
> Thank you for this question.
>
> First, as shown in Fig. 7 in Appendix E.1, the prompt asks the LLM to generate descriptions with a "hint" (an entity description from the original KG dataset). This helps differentiate entities with the same name during this step. In our paper, only YAGO3-10 does not have such descriptions, but it does not have this issue.
>
> Second, we can also mitigate this by using strategies like MPIKGC/CD (mentioned in W1.2). Feeding the LLM triples of entities from the training set helps provide context and differentiate entities within the KG.

---

> ### Author Response · Authors · 2024-08-13
>
> Dear Reviewer hiNn,
>
> We greatly appreciate your thoughtful review and the time you have taken to provide detailed feedback on our work. In our author response, we have addressed your valuable comments regarding related work, motivation, and the several technical aspects of KG-FIT.
>
> As **the discussion period nears its end (in 24 hours)**, we would be grateful if you could take a moment to review our response and let us know if there are any remaining concerns or if our clarifications have adequately addressed your points.
>
> Thank you once again for your efforts in evaluating our submission.
>
> Sincerely,
>
> The Authors

---

### Author Rebuttal · Authors · 2024-08-06

Dear Reviewers,

We sincerely thank you for your thoughtful and constructive feedback on our submission "Knowledge Graph Fine-Tuning Upon Open-World Knowledge from Large Language Models". We deeply appreciate the time and effort you've invested in reviewing our work.

> **[Strengths of Our Work]**

We are grateful that multiple reviewers recognized several strengths of our work:
1. The effectiveness of KG-FIT in leveraging knowledge from LLMs to significantly enhance KG embeddings for the link prediction task (All Reviewers).
2. The extensive experimental evaluation, including comparisons with numerous baselines across multiple datasets and KG embedding backbones (Reviewers *G3dr*, *dKZ2*, and *9oop*).
3. The clear presentation, including well-structured paper organization and clear figures (Reviewers *hiNn*, *G3dr*, and *9oop*).
4. The provision of code for reproducibility and detailed experimental setup descriptions (Reviewers *G3dr* and *9oop*).

> **[Our Responses to Weaknesses]**

We acknowledge the concerns raised and have addressed them in our **individual rebuttals**. We have also attached a **Rebuttal PDF** which includes several tables and a figure showing new experimental results to address reviewers' concerns. Here's an overview of its content and key findings:

1. Table A: Performance of KG-FIT augmented by LLM-based KG textual information enhancement methods (MPIKGC and CD) on FB15K-237.
   * Key finding: KG-FIT can be further enhanced by incorporating improved textual information, showing consistent improvements across all metrics.

2. Table B: Comparison with additional LLM-based baselines (KG-LLM and KICGPT) on FB15K-237 and WN18RR.
   * Key finding: KG-FIT outperforms KG-LLM and significantly enhances the performance of KICGPT when used as its backbone retriever, demonstrating its effectiveness in combining with LLM-based methods.

3. Figure A: Sensitivity analysis demonstrating KG-FIT-HAKE's robustness to hyperparameter variations.
   * Key finding: KG-FIT maintains stable performance across various hyperparameter settings, indicating its robustness and potential for easy adaptation to new datasets.

4. Table C: Comparison with additional recent KG embedding baselines (CompoundE, GIE, and DualE) across FB15K-237, YAGO3-10, and PrimeKG.
   * Key finding: KG-FIT consistently enhances the performance of these state-of-the-art KG embedding methods across all datasets, achieving new state-of-the-art results.

We have provided detailed, point-by-point responses to each reviewer's specific concerns in our individual rebuttals. We encourage you to refer to these for in-depth discussions on particular aspects of our work. We are committed to incorporating your valuable feedback to further improve our paper.

Thank you again for your time and expertise in reviewing our submission.

---

### Decision · Program_Chairs · 2024-09-25

**Decision:**

Accept (poster)

**Comment:**

This is a paper on KG embeddings, where entity descriptions from LLMs are used to improve the embedding training. The paper yielded 4, 5, 5, 6. One reviewer who gave a 4 rating did not engage in discussion, whereas another later increased their score to 5. I have looked at author response for both these and other reviewers. Authors have provided additional experiments to provide good evidence for issues raised (e.g., comparison of other LLM-based enhancement approaches, comparison with other LLM-based baselines, experiments over more recent models, other clarifications especially regarding why do the research at all and rather simply use LLMs)

The positive reviewer also has interesting points such as what if LLM doesnt know these entities. Authors give some high level response to this question, but do not perform any experiments on the robustness of the model in such a case. This is an important question, and some experiments may have benefited the paper. Unfortunately, none(?) of the previous papers that combine LLMs with structure have done this experiment either... so while interesting, should not become the sole reason for rejection.

Finally, one criticism which is not addressed adequately is that some other papers like NBFNet (and more recently this https://arxiv.org/pdf/2311.03780) combine GNNs with LLMs, and have excellent performance on one of the hard datasets: (FB15K-237, 43 MRR, 33 Hits@1). The authors' results are much weaker. This is because the authors combine static embedding based models and not GNN-architectures that have additional power because they compute dynamic embeddings for a node in a query dependent way.  Authors relegate this combination (of their approach with GNNs) to future work.

Overall, this is a paper where there aren't that many things wrong. Its a decent new idea, and good extensive experiments on a subset of models, leading to improvements compared to those models... not always generating a state of the art performance. I personally do not believe that a paper must only produce state of the art performance to get accepted. The idea of creating a hierarchy of entities and using that to train KGs is somewhat interesting.

Is the paper groundbreaking? No. But, it will be a decent addition to NeurIPS if there is space.